# Four New Species of *Russula* Subsection *Sardoninae* from China

**DOI:** 10.3390/jof9020199

**Published:** 2023-02-03

**Authors:** Guo-Jie Li, Tie-Zhi Liu, Shou-Mian Li, Shi-Yi Zhao, Cai-Yun Niu, Zhen-Zhen Liu, Xue-Jiao Xie, Xu Zhang, Lu-Yao Shi, Yao-Bin Guo, Ke Wang, Bin Cao, Rui-Lin Zhao, Ming Li, Chun-Ying Deng, Tie-Zheng Wei

**Affiliations:** 1Key Laboratory of Vegetable Germplasm Innovation and Utilization of Hebei, Collaborative Innovation Center of Vegetable, College of Horticulture, Hebei Agricultural University, Baoding 071001, China; 2College of Chemistry and Life Sciences, Chifeng University, Chifeng 024000, China; 3State Key Laboratory of Mycology, Institute of Microbiology, Chinese Academy of Sciences, Beijing 100101, China; 4Guizhou Institute of Biology, Guizhou Academy of Sciences, Guiyang 550009, China

**Keywords:** *Basidiomycota*, marcofungus, phylogeny, *Russulaceae*, taxonomy

## Abstract

Four new species of *Russula* subsection *Sardoninae* from northern and southwestern China under coniferous and deciduous trees are proposed as *R. begonia*, *R. photinia*, *R. rhodochroa*, and *R. rufa*. Illustrations and descriptions of *R. gracillima*, *R. leucomarginata*, *R. roseola*, and the above four new species are provided based on evidence of morphological characters and phylogenetic analyses of the internal transcribed spacer (ITS), as well as the multi-locus of mtSSU, nLSU, *rpb1*, *rpb2* and *tef1*-α. The relationships between these new species and allied taxa are discussed.

## 1. Introduction

*Russula* Pers. is an ectomycorrhizal genus of high species diversity which is commonly distributed throughout boreal, temperate and tropical ecosystems worldwide [1,2,3]. It often appears as one of the main ectomycorrhizal macrofungal groups in forest and shrubs [4,5,6]. This group, which has been estimated at at least 2000 species worldwide, plays a significant role in preserving the biodiversity of forest ecosystems. The symbiotic association of ectomycorrhiza established by Russulas and their host plants is able to improve the absorption of nutrients including nitrogen, water and minerals, which are helpful in overcoming barriers caused by adverse conditions [7,8]. Anatomical features of ectomycorrhiza are the conserved, morphological, infrageneric taxonomic bases for *Russula* [2]. The developed ectomycorrhizal network of *Russula* also breeds quantities of fruiting bodies, some of which are regarded as edible mushrooms with considerable regional economic value [9,10,11,12,13]. Taxonomy of *Russula* are based on macro-morphological characters in the early stage, such as pileus surface tinge, context taste, and spore print colour. Then, micro-morphological characters are introduced as taxonomic evidence of this genus, e.g., the shape and size of hymenial and pileipellis elements. The phylogenetic topology of certain DNA regions have been regarded as an important taxonomic basis in recent decades. The genes commonly analysed in *Russula* taxonomy are ITS, mtSSU, nLSU, *rpb1*, *rpb2* and *tef1*-α [14,15].

The fairly large basidiomata, brightly coloured pileus, brittle context, and absence of annuluses or volva make *Russula* species distinguishable in the field. The *Russula* subsection *Sardoninae* Singer, a member of *Russula* section *Russula* Pers., is a representative group of the reddish-capped species of the genus [14]. This subsection is characterized by pink-to-lilac tinged stipes, adnate or subdecurrent lamellae, stiff, acrid tasted context, cream-to-ocher basidiospore print, and non-crusted pileocystidia in pileipellis [14,15]. Four series, *Exalbicans*, *Sardonia*, *Persicina* and *Sanguinea*, were further established under this subsection based on pileus colour and ectomycorrhizal symbiotic habitat [14]. The independence of this subsection was also supported by ITS and the multi-gene phylogeny of the *Russula* genus [2,16,17]. Most of the known *Russula* subsection *Sardoninae* species were identified in Europe and North America [14,15,18,19,20,21,22], *Russula choptae* A. Ghosh and K. Das, *R. leucomarginata* B. Chen et al., *R. subsanguinaria* B. Chen et al., *R. roseola* B. Chen et al. and *R. thindii* K. Das and S.L. Mill. originally reported based on materials from Asia [23,24,25]. Among the presences of *R*. subsection *Sardoninae* species reported in China, namely, *R. cavipes* Britzelm., *R. exalbicans* (Pers.) Melzer and Zvára, *R. gracillima* Jul. Schäff., *R. leucomarginata*, *R. luteotacta* Rea, *R. persicina* Krombh., *R. roseola*, *R. sardonia* Fr., *R. sanguinea* Fr., *R. subsanguinaria* and *R. thindii*, only species identified in Asia have been supported by molecular phylogeny [17,25,26,27,28]. During recent surveys of *Russula* species in China, four members of *R*. subsection *Sardoninae*—*R. begonia*, *R. photinia*, *R. rhodochroa*, and *R. rufa*—were found as new to science with the support of morphological and phylogenetic analyses. Detailed morphological descriptions of *R*. *gracillima*, *R*. *leucomarginata* and *R. roseola* are also provided herein based on recent collections.

## 2. Materials and Methods

### 2.1. Voucher Specimen Collection and Morphological Description

*Russula* subsection *Sardoninae* specimens newly collected by the authors of this study are from northeastern, northern and southwestern regions of China. These specimens were sampled during field fungal forays in recent decades. Macro-morphological characters were recorded based on fresh samples under daylight. Photographs were shot with Sony DSC-WX220/HX-400/NEX-5T and Nikon D750 digital cameras. Colour codes and names for macro-morphological characters followed those of Ridgway [29]. Spore-print colour codes followed those of Romagnesi [15]. Dehydration of fresh basidiomata to constant weight was performed using a Fruit LT-21 digital food drier at 55–65 °C for 12 h. Exsiccata were preserved in Mycological Herbarium of Chifeng University (CFSZ); Herbarium of Hebei Agricultural University (HBAU); and Herbarium of Mycology, Institute of Microbiology, Chinese Academy of Sciences (HMAS). Optical microscopic observations were carried out with a Nikon Eclipse Ci-L photon microscope. Pictures for line drawings were taken with a Cossim U3CMOS1400 camera which was fixed to the photon microscope. Tiny pieces of lamellae and pileipellis were taken from dried specimens and rehydrated in 5% KOH solution. Tissues were anticlinally sliced with a Flying Eagle razor blade and immersed in Congo Red solution for measurements and observations. Basidiospores ornamentations were observed in Melzer’s reagent. Sulphovanillin (SV) was used to test the contents and incrustations of hyphae and cystidia. Line drawings were created with a Wacom Intuos CTL6100WLK0 graphics drawing tablet. The apiculus and ornamentation of basidiospores and basidium sterigmata were excluded from microscopic measurement. The abbreviations a/b/c stand for measurements of c basidiospores from b basidiomata of a specimens. The size range of e–f in (d–) e–f (–g) means e–f covers 95% of the measured values, while d and g indicate extreme values. Plain and bold Q values represent the ratio of basidiospore length/width. Averages and standard deviations of Q values were calculated. The infrageneric taxonomic subdivision systems proposed by Sarnari [14] and Buyck et al. [2] were employed. Operating instructions not detailed above followed Yang [30] and Li [17].

### 2.2. DNA Extraction, PCR, and Sequencing

Genome DNA extractions were carried out from dried samples using the modified CTAB method protocol of Li [17]. The internal transcribed spacer (ITS) and 28S large subunit regions of ribosomal DNA were amplified with base primer pairs of ITS5/ITS4 and LROR/LR5 [31]. Partial 16S small subunit region of mitochondrial DNA was amplified with primer pair MS1/MS2 [31]. Partial largest and second largest RNA polymerase II regions (*rpb1* and *rpb2*) were amplified with primer pairs of rpb1-Ac/rpb1-Cr and brpb2-6f/frpb2-7cR [32,33,34]. Partial translation elongation factor 1-α (*tef1*-α) gene was amplified with primer pair EF-983F/EF-1567R [35]. PCR conditions in aforementioned references were applied. Amplification products were detected by 1.5% agarose gel electrophoresis. Then PCR product depurations were performed using a Biomed Nucleic Purification Kit. Sanger sequencing was performed with the same primers as for PCRs in a ThermoFisher ABI 3730XL DNA Analyzer by Biomed Gene Technology Company (Beijing, China). An ABI BigDye 3.1 Terminator Cycle Sequencing Kit was also applied in sequencing procedure. The newly acquired sequences in this analysis were deposited in GenBank (bold text in Appendix A).

### 2.3. Phylogenetic Analyses

The ITS sequences of representative and closely related species of *R.* subsection *Sardoninae* were selected from GenBank (https://www.ncbi.nlm.nih.gov, accessed on 18 November 2022) and UNITE databases (https://unite.ut.ee, accessed on 18 November 2022) referred to in previous analyses [1,16,17,19,20,21,23,24,25,28,36,37,38,39,40,41]. Referential sequences for multi-locus phylogenetic analyses were from Looney et al. [1], Buyck et al. [2], and Chen et al. [25]. Newly generated sequences were screened for identity through a BLAST in GenBank. Voucher specimens from this analysis are listed in Appendix A. Sequences were assembled and aligned in MAFFT 7.487 with E-INS-I strategy [42]. The matrix of dataset was manually adjusted with BioEdit 7.2.6 [43]. Multi-locus datasets were concatenated using SequenceMatrix v1.9 [44]. Species of *Russula* subsection *Atropurpurinae* Romagn, *R*. *atropurpurea* (Krombh.) Britzelm., *R. krombholzii* Shaffer, *R. ochracea* Fr. and *R. vinacea* Burl. were used as outgroups in ITS and multi-locus of phylogenetic analyses. Maximum likelihood (ML), maximum parsimony (MP), and Bayesian analyses were performed to clarify the ITS phylogenetic relationships between the new species and closely related known ones. Maximum likelihood and Bayesian analyses were performed for multi-locus phylogenetic analyses of mtSSU, nLSU, *rpb1*, *rpb2* and *tef1*-α regions. Maximum likelihood analyses were carried out in raxmlGUI 2.0.3 [45]. The number of bootstrap (BS) replications was set to 1000. Substitution models of GTR + I, GTR + G, and GTR + G + I were used in ML analyses. The Bayesian analysis (BI) was conducted in MrBayes 3.2.7 using Metropolis-coupled Markov-chain Monte Carlo (MCMCMC) methods [46,47]. MrModeltest 2.3 was used for the estimation of best-fit model with Akaike information criterion (AIC) [48] for Bayesian analyses. The four Markov-chain runs were set to 1.5 × 10^6^ generations. Trees were saved at frequency of every 100th generation. The standard deviation of split frequencies was stably kept below the 0.01 threshold when the running was over. The discard for burn-in phase of each analysis was the first 25% of sampling trees. The remaining 75% of trees were used for computation of Bayesian posterior probabilities (PPs), following a 50% majority-consensus rule. The convergences of runs were assessed using Tracer 1.7 [49]. MP analysis was performed in PAUP^*^ 4.01 [50]. Gaps in alignment were regarded as missing data. All sites in the matrix were treated as unordered and unweighted. The algorithm in heuristic search of MP analysis followed the tree bisection–reconstruction (TBR). The stabilities of clades were assessed through bootstrap analysis of 1000 replicates [51]. A Kishino–Hasegawa test (KH test) was carried out to determine the significances of tree differences [52]. The values of consistency index (CI), retention index (RI), rescaled consistency index (RC), homoplasy index (HI), and tree length (TL) were also calculated. Trees were depicted and polished in Figtree 1.4.4 (http://tree.bio.ed.ac.uk/, accessed on 18 November 2022).

## 3. Results

### 3.1. Phylogenetic Analyses

The GenBank BLAST searches of the ITS regions for the new species in this study returned the following matches: environmental samples of ectomycorrhizal fungi (EMF) from Northwest China (JF748110, JQ318655, KR149740) [53,54] and specimens from North China (MN737469; MW554106) have 99–100% identification with 96–99% coverages of *R. begonia*, European *R. persicina* specimens (UDB002502 and UDB015984) have 95% identification with 92–98% coverages of *R. begonia*; specimen of *R. subsanguinaria* from northeastern China (MW301625) [25] has 97% identification of coverage with *R. leucomarginata*; the Asian specimens of *R. thindii* from Himalayan regions of China and India (KM386693, KU290399) [23,28] best match with *R. roseola*: 97% identifications of 96–98% of coverages; *R. gracillima* specimens from Europe and northern China (KR364094, MW850414) ([55], this study) have the highest identification and coverages, 97% and 99%, with *R. photinia*; the *R. subsanguinaria* sample from Jilin, northeastern China (MW301621) [25] has 99% identification of 90% coverage with *R. rhodochroa*; for *R. rufa*, specimens from Europe and North America (AY061711, KX813591, FJ845434, HQ604841) [16,20,37] identified as *R. fragilis*, *R. queletii*, *R. salishensis* and *R. sanguinea* have 99% identification of 96% coverages.

The ITS sequence matrix of this analysis was 576 bp in length, representing 41 species. The dataset contains 182 bp of ITS-1, 169 bp of 5.8S and 212 bp of ITS-2. Of the 576 characters in the ITS phylogenetic analyses, 368 characters were constant, 35 variable characters were parsimony-uninformative, and 173 characters were parsimony-informative. The tree has a CI of 0.442, an RI of 0.865, an RC of 0.382, an HI of 0.558, and a TL of 638. For BI analysis of the ITS region, the best substitution model is GTR + I + G. The total 142 sequences, including 60 newly generated ones, corresponded to 24 species and six complexes of *R.* subsection *Sardoninae*, as well as four species of an out group from *Russula* section *Atropurpurinae.*

The multi-locus sequence matrix is 4067 bp in length, representing 15 species. The dataset contains 564 bp of mtSSU, 879 bp of nLSU, 1253 bp of *rpb1*, 776 bp of *rpb2*, and 595 bp of *tef*-1α. For BI analysis of the multi-locus region, the best substitution model is GTR + I + G for mtSSU, nLSU, *rpb1* and *rpb2*, and SYM + I + G for *tef*-1α. The total 211 sequences, including 136 newly generated ones, corresponded to 16 species of *R.* subsection *Sardoninae* and two species of an out-group species from *Russula* section *Atropurpurinae*.

Based on the basal rank consistency of phylogenetic topologies generated by BI, ML, and MP analyses, only ML trees are presented in Figure 1 and Figure 2. The resulting phylograms showed that clades of the four new species were all well-supported by multi-locus phylogenetic analyses. These new species can be significantly distinguished from known species (Figure 2). The independences of *R*. *begonia* and *R. photinia* are supported by ITS phylogeny. *Russula rufa* and *R*. *rhodochroa* cannot be clearly distinguished from closely related taxa in ITS phylogenetic analyses (Figure 1). The *Russula begonia* clade closely nests with that of *R*. *persicina* in ITS and multi-locus phylogenetic topologies (Figure 1 and Figure 2). The clade of *R. roseola* and *R. rufa* cluster with specimens of *R. americana* (Singer) Singer, *R. fuscorubroides* Bon, *R. queletii*, *R. salishensis*, *R. thindii*, *R. torulosa* and samples identified as *R. fragilis* and *R. sanguinea* with strong support from ITS phylogeny (MLBS 92, MPBS 91, PP 1, in Clade F, Figure 1). Multi-locus phylogenetic analyses also supported *R. roseola* and *R. rufa* have a close relationship with *R. queletii* (Figure 2). The clade of *R. photinia* clusters with that of *R. gracillima* in ITS phylogeny (MLBS 64, MPBS 78, in Clade G, Figure 1). Bootstrap and posterior probability values indicate *R. photinia*-*R. gracillima* clades are strongly supported by multi-locus phylogeny (Figure 2). The clade of the *R*. *rhodochroa-R. subsanguinaria* complex forms a passably supported clade with samplings of *R. leucomarginata*, *R. rhodocephala*, *R. sanguinea*, and two unnamed *Russula* species (MLBS 58, PP 1, in Clade F, Figure 1). The relationships between *R. leucomarginata*, *R. rhodochroa* and *R. subsanguinaria* are further supported by multi-locus phylogenetic analyses, whereas the topology of *R. rhodochroa* does not cluster together with *R. subsanguinaria* (Figure 2).

### 3.2. Taxonomy

***Russula begonia* G.J. Li, T.Z. Liu,** and **T.Z. Wei, sp. nov.** (Figure 3a–d, Figure 4a and Figure 5).


**Fungal Names: FN 571243**


**Etymology:** The specific epithet *begonia* refers to the pinkish red pileus, reminiscent of the flower colour of the plant genus *Begonia* L.

**Holotype:** China, Hebei Province, Baoding City, Fuping County, Tianshengqiao National Geological Park, in broad-leaved forest of *Quercus wutaishansea*; 38°51′53.5′′N, 113°51′35.1′′E, alt. 964 m; G.J. Li, Y.B. Guo, X. Zhang, X.J. Xie, T.T. Fan; 23 August 2020; 20200230 (HBAU15564).

**Diagnosis:** Basidiomata of a small-to-medium size. Pileus reddish tinged, intermixed with a rufous fringe, slightly glutinous and faintly striated at margin. Lamellae adnate, sometimes decurrent, white to cream, unchanging when bruised; edge even; lamellulae not observed. Stipe central, rarely subcentral, 32–74 × 11–15 mm, cylindrical to subcylindrical, white, often stained with pale reddish tinge. Context 2–5 mm thick at pileus centre, white, unchanging but rarely turning cream when bruised, taste somewhat acrid, odour indistinct. Spore print pale cream. Basidiospores (5.7–) 6.0–7.5 (–7.6) × (4.3–) 4.7–6.4 (–7.0) μm, 6.8 × 5.6 μm on average, subglobose to broadly ellipsoid, rarely globose and ellipsoid, ornamentations 0.4–1.0 μm in height, mostly linked as short to long crests and ridges forming an incomplete reticulum, often intermixed with isolated verrucae. Basidia 29–42 × 8–12 μm, subcylindrical to subclavate. *Cheliocystidia* 55–73 × 9–14 μm, subclavate to subcylindrical, at times clavate to fusiform, apex obtuse. *Pleurocystidia* dispersed, 58–75 × 9–14 μm, subclavate to clavate, rarely cylindrical, apex papilliform, rarely obtuse. *Pileipellis* composed of two layers. Suprapellis an ixotrichoderm at pileus centre, pileus margin a trichoderm. Pileocystidia distributed mainly in suprapellis, disperse at pileus margin, non-septated, sometimes one- to two-septated, cylindrical to subclavate, apex obtuse. Habitat in broad-leaved forests.

**Description:** Basidiomata small to medium sized. Pileus 27–68 mm in diam., first hemispherical, crateriform to pulvinate, rarely campaniform; then convex to flat when mature, at times centrically concave, dull, slightly glutinous when wet, reddish tinged, intermixed with rufous fringe, Light Jasper Red (XIII3′b), Light Coral Red (XIII5′b), sometimes Acajou Red (XIII1′i), Brick Red (XIII5′k) to Pompeian Red (XIII3′i) at centre when mature, partly turning Onion Skin Pink (XXVIII11′′b), Begonia Rose (I1b) to Japan Rose (XXVIII9′′b) when old and dry; margin acute, incurved when young, often undulate, straight at last, somewhat curled-up and cracked, faintly striated 1/6–1/5 from the edge inwards, peeling 1/4–1/3 towards the centre, with a reddish to pinkish tinge of Grenadine (II7b), Grenadine Pink (II7d) and Bittersweet Orange (II9b). Lamellae adnate, sometimes decurrent when mature, 3–5 mm in height at the midpoint of the pileus radius, fragile, infrequently forked near the stipe and edge, slightly interveined, White (LIII), a cream tinge of Light Buff (XV17′f) with age, unchanging when bruised; edge even, narrowing towards the margin, 10–15 blades per cm at the pileus margin; lamellulae not observed. Stipe central, rarely subcentral, 32–74 × 11–15 mm, cylindrical to subcylindrical, infrequently subclavate to clavate, slightly tapered towards the base, annulus absent, smooth when young, sometimes longitudinally rugulose with age, White (LIII), often stained with pale reddish tinge of Shrimp Pink (I5f), La France Pink (I3f), and Hermosa Pink (I1f) at middle and lower parts, stuffed when young, hollow at last. Context 2–5 mm thick at pileus centre, White (LIII), unchanging, rarely slowly turning pale cream tinge of Catridge Buff (XXX19′′f) when bruised, Cream Buff (XXX19′f) when old, fragile, taste somewhat acrid, turning mild when old, odour indistinct. Spore print pale cream (IIa–IIb).

Basidiospores [200/4/4] (5.7–) 6.0–7.5 (–7.6) × (4.3–) 4.7–6.4 (–7.0) μm, Q = (1.04–) 1.07–1.35 (–1.40) (Q = 1.21 ± 0.09), 6.8 × 5.6 μm in average, subglobose to broadly ellipsoid, rarely globose and ellipsoid, ornamentations composed of verrucous to subcylindrical, infrequently conical, moderately distant to dense ((5–)6–7 in a 3 μm diam. circle), amyloid warts 0.4–1.0 μm in height, mostly linked as short to long crests and ridges forming an incomplete reticulum, often intermixed with isolated verrucae, fused in short chains and pairs (1–3(–4) fusions in the circle), line connections absent or dispersed (0–2 line connections in the circle); suprahilar area amyloid and distict. *Basidia* 29–42 × 8–12 μm, subcylindrical to subclavate, rarely clavate and cylindrical, four-spored, hyaline; sterigmata 5–8 μm in length, straight to slightly tortuous. *Cheilocystidia* 55–73 × 9–14 μm, subclavate to subcylindrical, at times clavate to fusiform, apex obtuse, contents crystalline to granular, partly sparse, greyish in SV. Pleurocystidia dispersed, ca. 300/mm^2^, 58–75 × 9–14 μm, subclavate to clavate, rarely cylindrical, apex papilliform, rarely obtuse, contents granular to crystalline, greyish in SV. Subhymenium 40–60 μm thick, cellular to pseudoparenchymatous, composed of intertwined, inflated, rarely branched hyphae 7–11 μm in width. *Pileipellis* composed of two layers. *Suprapellis* 70–100 μm thick, an ixotrichoderm at pileus centre, composed of mostly erect, rarely branched, hyaline hyphae; primordial hyphae absent, terminal cells cylindrical, apex obtuse, rarely tapered, 15–35 μm in length; pileus margin a trichoderm, containing a certain number of repent-to-oblique elements, terminal cells ellipsoid, subcylindrical to clavate, often obtuse, inflated to lageniform at apex; subapical cells 2–5 septate, cylindrical to ellipsoid, rarely inflated. *Pileocystidia* distributed mainly in suprapellis, disperse at pileus margin, non-septated, sometimes one- to two-septated, cylindrical to subclavate, apex obtuse, 4–10 μm in width, often in suprapellis 40–60 μm in length, rarely up to 90 μm, apex obtuse, contents granular, dense, blackish in SV. Subpellis a cutis, composed of mostly repent elongated cylindrical hyphae, often intermixed with ellipsoid to irregular shaped cells 3–8 μm in width. Clamp connections not observed in all tissues.

**Habitat and distribution:** Single or scattered broad-leaved forests.

**The rest of the specimens examined:** China, Hebei Province, Baoding City, Fuping County, Tianshengqiao National Geological Park; 38°52′17.3′′N, 113°51′43.2″E; alt. 1125 m; in broad-leaved forest of *Quercus wutaishansea*; G.J. Li, Y.B. Guo, X. Zhang, X.J. Xie, T.T. Fan, 23 August 2020, 20200232 (HBAU15565); Inner Mongolia Autonomous Region, Chifeng City, Ningcheng County, Heilihe National Nature Reserve, Sidaogou Township; 41°30′39.2′′N, 118°23′53.4′′E; in mixed forest of *Quercus mongolica*, Y.H. Shen, Y. Zhu, 20 July 2004, (CFSZ2192); ibid, Sandaohe Village, 41°131′17.3′′N, 118°18′33.7′′E; T.Z. Liu; alt. 1054 m; Y.H. Tan; 23 August 2018 (CFSZ20023).

**Note:** The closely related *R. persicina* differs from this new species in larger basidomata up to 12 cm in pileus diameter, strongly acrid-tasting context, longer basidia up to 52 μm in length, longer and slender hymenial cystidia 70–150 × 7–11 μm, and none or sparsely septate pileocystidia [14,15]. The other similar species, *R. luteotacta* Rea, can be distinguished by its lamellae and stipes turning yellow when injured, white spore print (Ia–Ib), larger basidiospores 7–9 × 5.7–7.5 µm, and longer and slender hymenial cystidia 67–100 × 5.7–7.5(11) µm [14,15]. *Russula* subsect. *Sardoninae* members distributed in China differ in the following morphological characters: *R. exalbicans* has a context which turns grey when moist, an ochreous spore print (IIIa–IIIb), larger basidiospores 7–9.6 × 5.4–7 µm, longer basidia up to 52 µm in length, and a habitat of *Betula* forest. *R. gracillima* has a versi-coloured, sometimes greenish, dirty-violet pileus surface; larger basidiospores of 7.2–8.5 × 5.2–6.5 µm; and longer hymenial cystidia up to 100 µm in length. *R. queletii* has olive-brown or greenish blotches in the pileus centre, a strongly acrid context with an obvious fruity smell, lamellae which rarely turn green, higher basidiospre ornamentations of 0.8–1.2 µm, and a habitat of the calcareous soil of *Picea* forest. *R. thindii* has a reddish to vinaceous stipe, pale yellowish basidiospore print, broadly ellipsoid to ellipsoid (Q = 1.16–1.32) basidiospores, and narrower pileocystidia of up to 7 µm in width. *R. subsanguinaria* has wider basidia of (25.4–)33–41.2(–42.6) × (5.6–)11–15.7(–17.7) µm; narrower hymenial cystidia of (37.7–)41–70.5(–98) × (4.9–)6–11.4(–16.7) µm; and a habitat of coniferous forest dominated by *pinus* spp. [14,15,23,25,28].

***Russula gracillima* Jul. Schäff., Z. Pilzk. 10: 105. (1931)** (Figure 4b, Figure 6a–d,g–h and Figure 7).

**Description:** Basidiomata small to medium sized. Pileus 25–82 mm in diam., flat-hemispheric to convex when young which later become planate, sometimes shallow-acetabuliform, reddish to pinkish tinged, intermixed with reddish-brown fringe, glabrous, Ox-Blood Red (I1k), Sanford’s Brown (II11k) at centre, sometimes faded to a pinkish tinge of Begonia Rose (I1b) to Rose Doree (I3b) when mature; margin acute to subacute, initially incurved, outstretched when mature, sometimes corrugated or uplifted, rarely cracked, striate 1/6–1/5 from the edge inwards, peeling 1/5–1/3 towards the centre, with a pinkish tinge of Hermosa Pink (I1f), La France Pink (I3f) to Venetian Pink (XIII1′f). Lamellae adnate to adnexed, 2–4 mm in height at midway point of pileus radius, fragile, often forked near the stipe, slightly interveined, white to pale cream, first White (LIII), unchanging when bruised, turning Ivory Yellow (XXX21′′f) to Cartridge Buff (XXX19′′f) with age; edge even, tapered towards the margin, 8–15 pieces at 1 cm near the pileus margin, lamellulae rare. Stipe central to slightly subcentral, 36–107 × 8–15 mm, subcylindrical to subclavate, sometimes cylindrical, slightly tapered towards the upper parts, annulus absent, smooth when young, often longitudinally rugulose at last, most of the surface pale red to pink of Shrimp Pink (I5f), Safrano Pink (II7f), and Orient Pink (II9f) except for a White (LIII) tinge towards the top, solid when young, then fistulous, then hollow. Context 2–5 mm thick at pileus centre, White (LIII) first, unchanging when bruised, slowly becoming Maize Yellow (III19f) to Martius Yellow (III23f) with age, brittle, taste slightly acrid, mild when old, odour indistinct. Spore print pale cream (IIa–IIb).

Basidiospores (300/6/6) (6.2–) 6.7–8.5 (–8.8) × (5.0–) 5.5–7.2 (–7.5) μm, Q = (1.03–) 1.08–1.39 (–1.35) (Q = 1.22 ± 0.08), 7.7 × 6.3 μm on average, subglobose, broadly ellipsoid to ellipsoid, rarely globose, ornamentation amyloid, composed of verrucae often connected as short crests and ridges of 0.5–1.1 μm in height, dense (6–9(–11) in a 3 μm diam. circle), not forming a complete network, reticulate, often fused in long chains (5–9(–11) fushions in the circle), rarely connected with line connections (1–4 connections in the circle); suprahilar area amyloid and distict. Basidia 28–46 × 8–12 μm, clavate to subclavate, four-spored, hyaline; sterigmata 3–7 μm in length, slight curving. Cheilocystidia 56–77 × 9–14 μm, clavate to subclavate, rarely subcylindrical, apex obtuse, contents granular, greyish in SV. Pleurocystidia moderately numerous, ca. 1000–1500/mm^2^, 51–80 × 9–14 μm, clavate to subclavate, apex obtuse, rarely subacute, contents granular, greyish in SV. Subhymenium 60–70 μm thick; pseudoparenchymatous; composed of thin-walled, ellipsoid-to-subglobose, rarely irregular hyphae 8–13 μm in width. Pileipellis composed of suprapellis and subpellis, not sharply delimited from underlying sphaerocytes in context. Suprapellis 80–120 μm thick; a trichoderm at pileus centre; composed of hyaline, septate, repent and vertical hyphae; rarely branched; primordial hyphae absent; terminal cells 15–25 × 4–8 μm, cylindrical to ventricose; apex obtuse; subapical cells cylindrical, sometimes inflated, barrel-shaped to broadly ellipsoid, even subglobose; compared with pileus centre, the suprapellis of pileus margin contains more inclined elements, 3–7 μm in width, inflated hyphal cells rare; pileocystidia 30–80 × 5–8 μm, cylindrical, often 1–4 septate, rarely multi-septate, apex obtuse, contents granulate, mostly dense, partly sparse, blackish grey in SV. Subpellis a cutis, 100–160 μm thick, composed of mostly horizontal, often inflated, loosely intricate, hyaline hyphae 3–9 μm in width. Clamp connections absent in all tissues.

**Habitat and distribution:** Single or scattered in coniferous and broad-leaved mixed forests.

**Specimens examined:** China, Hebei Province, Shijiazhuang City, Pingshan County, Hehekou Township, Tuoliang National Nature Reserve, in mixed forest of *Larix principis-rupprechtii*, *Quercus liaotungensis*, and *Q. mongolica*; G.J. Li, Y.B. Guo, X. Zhang, X.J. Xie, T.T. Fan; 38°43′57.5′′N, 113°49′34.3′′E; 23 August 2020; 20200097 (HBAU15484); ibid, 23 August 2020, 20200175 (HBAU15527); ibid, Baoding City, Fuping County, Tianshengqiao National Geological Park, in mixed forest of *Pinus tabuliformis* and *Quercus wutaishansea*; G.J. Li, Y.B. Guo; 18 August 2019, 20190267 (HBAU15080); Inner Mongolia Autonomous Region, Chifeng City, Bairin Youqi, Saihanwula National Natural Reserve, Rongsheng Scenic Area; T.Z. Liu, C. Sun, J. Zhang, 19 July 2008, (CFSZ3399); ibid, Tiewangfengou Scenic Area, 12 September 2016, T.Z. Liu, Z.L. Song (CFSZ12238); Jilin Province, Changbaishan Protective Development Region, Chibei District, Erdaobaihe Township, X. Sun, Y. Zheng, G.J. Li, 12 August 2009, TM253 (HMAS261129); ibid, unknown collection date and site, anonymous collector (HMJAU5844).

Note: The pileus surface of Asian collections contains mainly reddish tinges for this, one of the most conspicuous macroscopic characters. The pileus colours of the European collections are more complicated. Pale green, dirty violet, and, rarely, citrine tinges are also present on pileus. The European specimens differ in several characteristics, such as the following: basidiospore ornamentations composed of mostly isolated warts with few connections, hymenial cystidia prolonged into a voluminous attenuated or lanceolate apex, and a habitat of the *Betula* forest [14,15]. The morphological and phylogenetic analyses of this study reveal a vast distribution of *R. gracillima* in China. The geographic localities of these samplings are from northern and northeastern parts of China. The flora of these known distribution areas are mainly Holarctic. It should be noted that the Asian distribution of this species was once reported as *R. gracilis* subsp. *altaica* Singer based on specimens of the Russian Altai Mountains. This taxon now has been regarded as a synonym of *R. gracillima* in subsequent studies. It has lower basidiospore ornamentations 0.2–0.5 μm in height [14,15]. As the Altai Mountains lie at the midpoint between Europe and East Asia, molecular phylogenetic analyses of *R. gracillima* in this region are needed to reveal its actual distribution in further studies. There are four collections from southwestern China (HMAS274621, HMAS274614, HMAS156221, and HMAS156227) nested with *R. gracillima* specimens in ITS phylogenetic analyses (Figure 1). The flora of ectomycorrhizal symbiotic plants for these four collections are different from the Holarctic flora for that of *R. gracillima*. HMAS274621 and HMAS274614 are collected from Hengduan Mountain region of Sino-Himalayan floral subkingdom. HMAS156221 and HMAS156227 are sampled from Yunnan–Myanmar–Thailand region of the Malay floral subkingdom. These specimens may represent some cryptic species; thus, they are not identified as *R. gracillima* in this study. Intensive morphological and multi-locus phylogenetic analyses are necessary for these samplings in future. *Russula subsanguinaria* can be distinguished from *R. gracillima* by the presence of brightly reddish tinged pileus, wider basidia of (25.4–)33–41.2(–42.6) × (5.6–)11–15.7(–17.7) µm, narrower hymenial cystidia of (37.7–)41–70.5(–98) × (4.9–)6–11.4(–16.7) µm, and a habitat of coniferous forest dominated by *Pinus* spp. [25].

***Russula leucomarginata* B. Chen, J.F. Liang, and X.M. Jiang, *Life* 12 (4, no. 480): 8. (2022)** Figure 4f and Figure 8).

**Description:** Basidiomata medium sized. Pileus 33–65 mm in diam., flat-hemispheric, umbonate to convex when young, turning planate at last, sometimes centrally concave, brightly reddish tinged, Carmine (I1i), Nopal Red (I3i), and Brazil Red (I5i) at the centre, occasionally faded to English Red (II7i) to Orange-Rufous (II11i), turning Sanford’s Brown (II11k), Pecan Brown (XXVIII11′′i) to Cacao Brown (XXVIII9′′i) when mature, glossy, often glutinous when wet; margin acute, slightly incurved when young, the flat, occasionally wavy or uplifted, often cracked, non-striate, rarely indistinctly striate 1/6–1/5 from the edge inwards, peeling 1/6–1/4 of the radius, often with a reddish tinge of Testaceous (XXVIII9′′), Terra Cotta (XXVIII7′′), and Scarlet (I5). Lamellae adnate to adnexed, 2–4 mm in height at midway of the disc radius, occasionally forked near the stipe attachment, slightly interveined, White (LIII), unchanging when bruised, slowly becoming the pale cream tinge of Cartridge Buff (XXX19′′f), Cream Color (XVI19′f) to Maize Yellow (III19f) when old; edge even, tapering towards the margin, crowded, 8–15 blades at 1 cm from the pileus margin, lamellulae rare. Stipe central, 41–79 × 9–17 mm, cylindrical, rarely clavate to subclavate, sometimes ventricose towards the base; annulus absent, smooth first, longitudinally rugulose when mature, surface mostly reddish fringed of Scarlet Red (I3), Nopal Red (I3i), and Spectrum Red (I1), often faded with pinkish tinges of Geranium Pink (I3d), Eosine Pink (I1d), and Strawberry Pink (I5d) towards the upper parts, initially stuffed, turning fistulous to hollow when mature. Context 2–4 mm thick at centre of disc, White (LIII) first, unchanging, slowly becoming pale cream fringe of Maize Yellow (III19f) to Cream Color (XVI19′f) when old, brittle, taste more or less acrid, even mild when old, odour indistinct. Spore print pale cream (Romagnesi IIa–IIb).

Basidiospores (300/6/6) (5.5–) 5.8–8.1 (–8.6) × (4.5–) 4.9–7.4 (–6.9) μm, Q = (1.03–) 1.07–1.32 (–1.36) (Q = 1.20 ± 0.07), 7.0 × 5.8 μm in average, subglobose to broadly ellipsoid, rarely ellipsoid, ornamentations amyloid, composed of verrucose to conical bulges 0.2–1.0 μm in height, distant to moderately distant ((4–)5–6 in a 3 µm circle), partly linked, not forming a complete reticulum, fused in pairs and short lines (1–3 fusions in the circle), line connections dispersed (0–1 in the circle); suprahilar area disctinct, verruculose and amyloid. Basidia 30–41 × 9–13 μm, clavate to subclavate, rarely subcylindrical, four-spored, sometimes two-spored, hyaline; sterigmata 5–7 μm in length, often straight, at times slightly tortuous. Cheilocystidia 54–83 × 9–13 μm, clavate to fusiform, rarely subclavate, apex acuminate, often bluntly acuminate, moniliform, mucronate to lanceolate, contents sparsely distributed, granular, greyish in SV. Pleurocystidia dispersed to moderately numerous, ca. 700–1300/mm^2^, 50–85 × 8–13 μm, subfusiform to subclavate, rarely fushiform and subcylindrical, apex bluntly acuminate, sometimes moniliform or lanceolate, contents scattered, granular, greyish in SV. Subhymenium 30–40 μm thick, pseudoparenchymatous, not well-developed, composed of ellipsoid to subglobose cells 8–11 μm in width. Pileipellis composed of two layers, sharply distinguished from the sphaerocytes beneath. Suprapellis 80–110 μm thick, a trichoderm at pileus centre, composed of slantly oriented, non-gelatinized hyphae, primordial hyphae absent, terminal cells cylindrical, 14–25 × 3–7 μm, apex obtuse, subapical cells cylindrical, rarely branched or inflated, a trichoderm at pileus margin, mostly horizontal to ascending, rarely vertical, terminal cells 15–35 × 3–7 μm, cylindrical, obtuse at apex. Pileocystidia abundant in pileus margin, often embedded from subpellis in pileus centre, long, cylindrical, multi-septate, 5–13 μm in width, apex obtuse, rarely constricted, contents granulate, partly sparse, greyish in SV. Subpellis a cutis, 150–230 μm thick, composed of mostly repent, cylindrical to inflated hyphae 4–11 μm in width. Clamp connections absent in all tissues.

**Habitat and distribution:** Scattered, rarely solitary, in broad-leaved forests.

**Specimens examined:** China, Xizang Autonomous Region, Nyingchi City, Bomi County, Zhuolong Valley, in coniferous forest of *Picea likiangensis* var. *linzhiensis*, *Pinus armandii* and *Pinus densata*; T.Z Wei, Z.X. Wu, L. Yang, H.D. Zheng, X.C. Wang; 19 September 2016; 7682 (HMAS277592); ibid, 18 September 2016, 7543 (HMAS277405); ibid 7545 (HMAS277406); ibid, 419 (HMAS278669); ibid 7560 (HMAS277417); 7550 (HMAS277410).

**Note:** The morphology of the above collections is in-accordance with that of Chen et al. [25], except the latter has longer and wider hymenial cystidia (46.3–)58.3–79.2–100(–123) × (8.6–)10.2–12.4–14.6(–18.5) μm and narrower pileocystidia (3.7–)4.8–5.9–7(–8.6) μm. This species is close to *R. rhodocephala* and *R. sanguinea* in phylogenetic analyses. All of these species share brightly reddish-tinged pileus, pale reddish-to-pinkish tinged stipes, and a habitat of coniferous forest. *Russula rhodocephala* can be distinguished from larger basidiomata by its pileus diameter of up to 120 mm, mostly broadly ellipsoid to ellipsoid basidiospores (Q = (1–)1.24–1.25–1.26(–1.5)), shorter and narrower hymenial cystidia of 60–65 × 7–8 μm with an obtuse apex, and narrower pileocystidia up to 8 μm in width [20]. *Russula sanguinea* differs in its longer basidia of 40–57 × 9.7–11.7 µm, wider hymenial cystidia of 60–150 × 8.5–18.5 µm, and narrower pileocystidia of 4–8 µm in width [14,15]. The other two similar species of *R.* subsection *Sardoninae* growing in coniferous forest are *R. helodes* Melzer and *R. rhodopus* Zvára. *Russula helodes* has a distinct context which turns grey, an ochreous spore print (IIIa–IIIb), and spore ornamentations composed of a network of thin lines. *Russula rhodopus* can be distinguished from its longer basidia of up to 50 µm in length, and basidiospore reticulate, lower ornamentations of up to 0.6 µm in height, and longer hymenial cystidia of up to 100 µm in length [14].

***Russula photinia* G.J. Li, T.Z. Liu, and T.Z. Wei, sp. nov.** (Figure 4d,e, Figure 6e,f,i,j and Figure 9).


**Fungal Names: FN 571244**


**Etymology:** The specific epithet *photinia* refers to the reddish-tinged pileus with the colours similar to young leaves of the plant genus *Photinia* Lindl.

**Holotype:** China, Hebei Province, Shijiazhuang City, Pingshan County, Hehekou Township, Tuoliang National Nature Reserve, in broad-leaved forest of *Quercus mongolica*; G.J. Li, Y.B. Guo, X. Zhang, X.J. Xie, T.T. Fan; 22 August 2020; 20200186 (HBAU15536).

**Diagnosis:** *Basidiomata* medium sized. Pileus reddish tinged, intermixed with chocolate fringe, often partly faded to pale yellowish tinge, margin indistinctly striate. Lamellae adnate, White (LIII), unchanging when injured, gradually turning pale cream, lamellulae rare. Stipe 36–107 × 8–15 mm, clavate to subclavate, white, often flushed with pinkish-to-pale-reddish fringes. Context white, unchanging when bruised, slowly turning cream when mature, taste somewhat acrid, mild when old, odour indistinct. Spore print pale cream. Basidiospores (5.6–) 6.0–6.4 (–6.8) × (4.6–) 4.9–6.5 (–6.8) μm, 6.9 × 5.7 μm on average, subglobose, broadly ellipsoid to ellipsoid, occasionally globose, ornamentations 0.3–0.8 μm in height, composed of blunt to subconical verrucae of mostly interconnected fine lines. Basidia 34–50 × 8–12 μm, often clavate. *Cheilocystidia* 34–50 × 9–12 μm, clavate to subclavate. *Pleurocystidia* 32–50 × 9–12 μm, clavate, subclavate to fusiform. *Pileipellis* two-layered, suprapellis a trichoderm in pileus centre and margin. *Pileocystidia* dispersed, cylindrical, often with 1–4 septa. Habitat in broad-leaved forests.

**Description:** *Basidiomata* medium sized. Pileus 35–58 mm in diam.; initially pulvinate to hemispheric, becoming convex, plano-convex and planate when mature; rarely depressed in disc; reddish tinged; intermixed with chocolate fringe; glabrous; viscid to glutinous when humid; Orange-Citrine (IV19k), Antique Brown (III17k) to Sudan Brown (III15k) at centre, often partly faded to pale yellowish tinge of Lemon Yellow (IV23), Primrose Yellow (XXX23′′d) to Reed Yellow (XXX23′′b) and sometimes Etruscan Red (XXVII5′′), Ocher Red (XXXVII5′′b) to Dark Vinaceous (XXVII1′′) when mature; margin subacute, incurved when young, then outstretched, sometimes corrugated or uplifted, rarely cracked, indistinctly striated 1/4–1/3 from the edge inwards, peeling 1/5–1/4 from the radius, often with pale reddish to pinkish tinges of La France Pink (I3f), Strawberry Pink (I5d) to Peach Red (I5b). Lamellae adnate, 2–4 mm in height at mid-point of the radius, occasionally forked near the attachment of stipe, infrequently interveined, white to pale cream, first White (LIII), unchanging when injured, gradually turning the pale cream tinge of Cartridge Yellow (XXX19′′f) and Ivory Yellow (XXX21′′f) when old; edge even, narrowing towards the margin, crowded, 9–14 pieces at 1 cm near the pileus margin, lamellulae rare. Stipe central to subcentral, 36–107 × 8–15 mm, clavate to subclavate, sometimes cylindrical, often narrowing towards the upper parts, annulus absent, smooth first, longitudinally rugulose when mature, most of the surface White (LIII), often flushed with pinkish-to-pale-reddish fringes of Safrano Pink (II7f), Orient Pink (II9f), and Flesh Pink (XIII5′f), initially farctate, then tubular to hollow. Context 2–4 mm thick at pileus centre, initially White (LIII), unchanging when bruised, slowly turning Maize Yellow (III19f) to Baryta Yellow (IV21f) with age, brittle, taste somewhat acrid, mild when old, odour indistinct. Spore print pale cream (IIa–IIb).

Basidiospores (300/6/6) (5.6–) 6.0–6.4 (–6.8) × (4.6–) 4.9–6.5 (–6.8) μm, Q = (1.04–) 1.07–1.34 (–1.42) (Q = 1.21 ± 0.08), 6.9 × 5.7 μm on average, subglobose, broadly ellipsoid to ellipsoid, occasionally globose, verrucose, amyloid moderately distant to dense ((5–)6–7(–8) in a 3 μm circle), forming an incomplete reticulum, occasionally to frequently fused in pairs and short chains (1–3(–4) fusions in the circle), line connections dispersed (0–1 in the circle); suprahilar area disctinct, verruculose and amyloid. *Basidia* 34–50 × 8–12 μm, often clavate, at times subclave to subcylindrical, rarely cylindrical, four-spored, hyaline; sterigmata 4–8 μm in length, slightly flexed, rarely straight. *Cheilocystidia* 34–50 × 9–12 μm, clavate to subclavate, at times cynlindrical to subcylindrical, apex obtuse, rarely capitate, contents granular, partly sparse, greyish in SV. Pleurocystidia dispersed to moderately numerous, ca. 800–1200/mm^2^, 32–50 × 9–12 μm, clavate, subclavate to fusiform, rarely subcylindrical, apex often papilliform, rarely mucronate to lanceolate, contents granular, greyish in SV. Subhymenium ca. 50 μm thick, composed of intertwined, elongated to irregularly inflated hyphal cells 9–12 μm in width. *Pileipellis* two-layered, composed of suprapellis (60–110 μm thick) and subpellis (120–180 μm thick), distinctly delimited from the underlying context. *Suprapellis* a trichoderm in pileus centre; composed of horizontal, oblique to erect, hyaline hyphae 3–7 μm in width; primordial hyphae absent; terminal cells cylindrical, sometimes inflated to ventricose; apex obtuse; subapical cells cylindrical, sometimes inflated, rarely branched; suprapellis a trichoderm in pileus margin, composed of more vertical elements 3–8 μm in width. *Pileocystidia* dispersed; cylindrical; often with 1–4 septa; 5–10 μm in width; apex obtuse, rarely ventricose; contents dense, granular, and black in SV. Subpellis composed of repent to slightly inclined, cylindrical, septate hyaline hyphae often inflated to ellipsoids, 3–8 μm in width. Clamp connections not observed in all tissues.

**Habitat and distribution**: Single or scattered in broad-leaved forests.

**The rest of specimens examined:** China, Hebei Province, Shijiazhuang City, Pingshan County, Hehekou Township, Tuoliang National Nature Reserve, in broad-leaved forest of *Betula platyphylla*, *Juglans mandshurica*, *Quercus mongolica*, and *Populus davidiana*; G.J. Li, Y.B. Guo, X. Zhang, X.J. Xie, T.T. Fan; 22 August 2020; 20200203 (HBAU15550); Inner Mongolia Autonomous Region, Chifeng City, Bairin Youqi, Saihanwula National Natural Reserve, Zhenggou Scenic Area, in broad-leaved forest of *Betula platyphylla* and *Quercus mongolica*; T.Z. Liu, H.M. Tian, C. Sun; 1 September 2008 (CFSZ3719); ibid, (CFSZ3739); ibid, Harqin Qi, Ma’anshan Forest Park; T.Z. Liu, Y.M. Gao; 2 September 2019 (CFSZ21445).

**Note:** This species is similar to *R. exalbicans* in its reddish-to-pinkish pileus usually distinctly fading and a habitat of the *Betula* forest, the latter differs in its greenish tinges intermixed on pileus surfaces, context turning greyish when wet, ochreous spore print (IIIa–IIIb), and larger basidiospores of 7–9.6 × 5.4–7 μm. Another species in *R.* subsect. *Sardoninae* whose pileus sometimes completely fade into citrine is *R. gracillima*; it can be distinguished from *R. photinia* by its greenish and dirty-violet tinges on pileus surfaces, longer hymenial cystidia of up to 100 μm in length, and higher ornamentations of up to 1.1 μm in height [14,15]. The red form of this species can be mistaken for *R. begonia* in the forest of northern China due to its reddish-to-pinkish tinged pileus and stipes, cream basidiospore print, and habitat of Fagaceae trees. *Russula begonia* differs in its having shorter basidia of 29–42 × 8–12 μm, longer hymenial cystidia of up to 75 μm in length, and an ixotrichoderm at the pileus centre. For those pure reddish-capped species in *R.* subsect. *Sardoninae*, *R. helodes* is different from *R. photinia* in its distinctly greyish context, ochreous spore print (Romagnesi IIIa–IIIb), and larger basidiospores of 8–10.5 × 6.6–8.2 μm with ornamentations forming a complete network; *R. luteotacta* differs in its conspicuous discolouring, sometimes even all white, very acrid tasting context, white spore print (Ia–Ib), and larger basidiospores 7–9 × 5.7–7.5 µm; *R. sanguinea* can be distinguished by its larger basidiospores of 7.2–9.6 × 6.3–7.4 µm with mostly isolated ornamentations, longer and wider hymenial cystidia of 60–150 (and more) × 8.5–18.5 µm, and a habitat of the *Pinus* forest; *R. rhodopus* is different from *R. photinia* in its having ochreous spore print (IIc–IIIb), basidiospores ornamentations subreticulate to reticulate, longer hymenial cystidia of up to 100 µm, and a habitat of coniferous forest [14,15].

***Russula rhodochroa* G.J. Li, and Chun Y. Deng, sp. nov.** (Figure 4g, Figure 10a–k and Figure 11).

**Fungal Names:** FN 571245

**Etymology:** named for its reddish-tinged pileus which has a colour similar to petals of the plant genus *Rhododendron* L.

**Holotype:** China, Guizhou Province, Qiandongnan Miao and Dong Autonomous Prefecture, Liping County, Nanzhu Forest Farm, in coniferous forest of *Pinus massoniana*; X.J. Xie, X. Zhang, J.P. Li, G.J. Li; 15 October 2020; 20201060 (HBAU15905).

**Diagnosis:** Basidiomata small to medium sized. Pileus reddish tinged, margin indistinctly striated. Lamellae adnate, white to cream, unchanging when injured, lamellulae present. Stipe 33–54 × 7–13 mm, subcylindrical to subclavate, white, partly flushed with faintly pinkish fringes. Context initially white, unchanging when burnished, infrequently accruing a pale-cream tinge with age, taste slightly acrid, odour indistinct. Spore print pale cream. *Basidiospores* (5.3–) 5.9–7.7 (–8.3) × (5.0–) 5.5–7.1 μm, 7.0 × 6.2 μm on average, globose, subglobose to broadly ellipsoid; ornamentations 0.3–1.0 μm in height, mostly isolated, partly interconnected by fine lines, not reticulate. *Basidia* 31–38 × 9–13 μm, clavate to subclavate. *Cheilocystidia* not observed. *Pleurocystidia* 48–72 × 8–13 μm, clavate to subclavate, rarely fusiform, apex subacute to bluntly acuminate to obtuse, rarely lanceolate. *Pileipellis* two-layered, *Suprapellis* a trichoderm at the pileus centre, an ixotrichoderm at the margin, pileocystidia cylindrical, multi-septate. Habitat in coniferous forests.

**Description:** *Basidiomata* small to medium sized. Pileus 22–47 mm in diam., hemispheric when young, then becoming convex to flat, sometimes crateriform to shallow infundibuliform, reddish tinged, Scarlet (I5), Scarlet Red (I3), and Nopal Red (I3i) at centre, sometimes Flame Scarlet (I9), Grenadian Red (II7), and Tawny (XV13′i) when mature, glossy, glutinous when wet; margin acute to subacute, first incurved, then planate, sometimes wavy, often cracked, indistinctly striated 1/8–1/6 from the edge inwards, peeling 1/6–1/4 of the radius, with a pale orangish-to-reddish tinge of Pale Flesh Color (XIV7′f), Safrano Pink (II7f) to Orient Pink (II9f). Lamellae adnate, 2–3 mm in height at mid-point of the radius, rarely forking near the stipe, slightly interveined, White (LIII), unchanging when injured, slowly acquiring a cream tinge Pale Orange-Yellow (III17f) to Pale Yellow-Orange (III15f) with age; edge even, tapering towards the margin, crowded, 11–17 pieces at 1 cm near the pileus margin, lamellulae present. Stipe central; 33–54 × 7–13 mm; subcylindrical to cubclavate; rarely clavate; somewhat tapered towards the pileus; smooth when young; then turning longitudinally rugulose; surface White (LIII), partly flushed with faintly pinkish fringes of Shrimp Pink (I5f), Flesh Pink (XIII5′f) to Chatenay Pink (XIII3′f); first solid, tubular when mature. Context 2–3 mm thick at centre of disc; initially White (LIII); unchanging when burnished; infrequently turning pale cream tinge of Naphthalene Yellow (XVI23′f), Massicot Yellow (XVI21′f), and Maize Yellow (III19f) slowly with age; brittle; taste slightly acrid; odour indistinct. Spore print pale cream (Romagnesi IIa–IIb).

Basidiospores (100/2/2) (5.3–) 5.9–7.7 (–8.3) × (5.0–) 5.5–7.1 μm, Q = 1.01–1.24 (–1.28) (Q = 1.12 ± 0.06), 7.0 × 6.2 μm on average, globose, subglobose to broadly ellipsoid, ornamentations amyloid, moderately distant to dense ((5–)6–7(–9) in a 3 μm diam. circle), cristulate, composed of subcylindrical to subconical verrucae 0.3–1.0 μm in height, mostly isolated, partly interconnected by fine lines, not forming a complete reticulum, isolated to occasionally fused (0–2(–3) fusion in a circle), line connections dispersed (0–1 in a circle); suprahilar area amyloid, verruculose and distinct. Basidia 31–38 × 9–13 μm, clavate to subclavate, occasionally subcylindrical, mostly four-spored, rarely two-spored, hyaline; sterigmata 4–7 μm in length, often straight, at times tortuous. Cheilocystidia not observed. Pleurocystidia widely dispersed, ca. 200–300/mm^2^, 48–72 × 8–13 μm, clavate to subclavate, rarely fusiform, apex subacute to bluntly acuminate to obtuse, rarely lanceolate, contents crystalline to granular, black in SV. Subhymenium 30–40 μm thick, pseudoparenchymatous, composed of intertwined, irregularly inflated hyphal cells 9–13 μm in width. Pileipellis composed of suprapellis (90–150 μm thick) and subpellis (80–140 μm thick). Suprapellis a trichoderm at pileus centre, composed of erect, ascending, hyaline hyphae, primordial hyphae absent, terminal cells 15–30 × 3–6 μm, cylindrical, apex obtuse to ventricose, subapical cells cylindrical, sometimes inflated, barrel-shaped to subglobose, rarely branched; suprapellis of pileus margin an ixotrichoderm, composed of gelatinized, contains more repent to oblique elements, 3–5 μm in width, infrequently ramified; pileocystidia occasionally fasciculate at pileus centre, 5–9 μm in width, cylindrical, multi-septate, sometimes embedded in subpellis, apex obtuse, contents granulate, dense, blackish grey in SV. Subpellis a cutis, composed of horizontal to slightly ascending, interleaved, cylindrical, often inflated to broadly ellipsoid, hyaline hyphae 3–8 μm in width. Clamp connections not observed in all tissues. 

**Habitat and distribution:** Scattered in coniferous forests.

**The rest of specimens examined:** China, Guizhou Province, Qiandongnan Miao and Dong Autonomous Prefecture, Liping County, Liping National Forest Park, in coniferous forest of *Pinus massoniana*; X.J. Xie, X. Zhang, J.P. Li, G.J. Li; 14 October 2020; 20201014 (HBAU15901), 20201053 (HBAU15902), 20201013 (HBAU15903); ibid, 20201015 (HBAU15906); ibid, Nanzhu Forest Farm, in mixed forest of *Castanopsis fargesii*, *C. tibetana*, *Fagus longipetiolata*, *Liquidamba formosana*, *Lithocarpus litseifolius* and *Pinus massoniana*; 15 October 2020; 20201079 (HBAU15904); Yunnan Province, Kunming County, Kunming City, Songming County, Pingding Mountain; H.A. Wen, M.X. Zhou, G.J. Li; 11 August 2006; 06229-1 (HMAS187102); ibid; 06229-3 (HMAS187103); 14 October 2020; Dali Bai Autonomous Prefecture, Dali City, Cang Mountains, alt. 2600 m; H.Y. Su; July 2009 (HMAS196539).

**Note:** This species is close to *R. helodes*, *R. rhodopus*, *R. leucomarginata*, *R. sanguinaria*, *R. subsanguinaria* and *R. thindii* in its having brightly reddish pileus, cream to ochre spore print, pinkish flushed stipes, acrid-tasting context, multi-septate pileocystidia without incrustation, and habitat of coniferous forest [14,23,25]. *Russula helodes* can be separated from *R*. *rhodochroa* by the presence of ocherous spore print (Romagnesi IIIa-IIIb), completely reticulated spore ornamentations, longer basidia of up to 58 μm in length, and a habitat of *Sphagnum*. *Russula rhodopus* can be distinguished from *R*. *rhodochroa* through its larger basidiospores of 7.9–9 × 6.3–7.2 μm with larger pileus of up to 105 μm in diam., lower ornamentations of up to 0.6 μm in height, and longer basidia and hymenial cystidia of up to 65 μm and 100 μm in length [14]. *Russula leucomarginata* differs in its combination of pileus with tawny tinges; yellowish basidiospore print; larger, rarely ellipsoid basidiospores (6.2–) 7.0–8.4 (–9.3) × (5.5–) 6.0–7.0(–7.7) μm; and longer and wider pleurocystidia (46.3–)58.3–100(–123) × (8.6–)10.2–14.6(–18.5) μm [25]. *Russula sanguinaria* is different from *R*. *rhodochroa* in the presence of robust basidiomata with pileus of up to 100 mm in diam., the fruity smell of its context, larger basidiospores of 7.5–9(–10) × 6.5–8.2 μm with ornamentations composed of mostly isolated warts, and longer and wider hymenial cystidia of 60–150 × 8.5–18.5 μm [14,15]. *Russula subsanguinaria* can be differentiated by its larger basidiospores (6.3–)7.0–7.6–8.3(–9.3) × (5–)5.5–6.2–7(–7.7) μm with higher ornamentations of 0.6–1(–1.2) μm in height, wider basidia of (5.6–)11–15.7(–17.7) μm, and one-celled pileocystidia [25]. *Russula thindii* is different from *R*. *rhodochroa* in its larger basidiospores of 7.6–9.9 × 6.1–7.8 μm with vinaceous tinges on the pileus surface, ornamentations composed mostly of isolated conical to spinoid warts, longer and wider pleurocystidia of 55–123 × 12–15 μm, and narrower pileocystidia of up to 7 μm in width [23].

***Russula roseola* B. Chen, J.F. Liang,** and **X.M. Jiang, *Life* 12(4, no. 480): 12. (2022)** (Figure 3e–j, Figure 4c and Figure 12).

**Description:** Medium sized. *Pileus* 33–56 mm in diam.; initially hemispheric; rarely campanulate, then plano-convex to umbonate; flat when mature; sometimes shallowly infundibuliform; brightly reddish tinged; glabrous; strongly glutinous when young and humid; Nopal Red (I3i), Carmine (I1i) to Brazil Red (I5i) at centre; Dragon’s Blood Red (XIII5′i), becoming reddish brown of Pompeian Red (XIII3′i) to Ferruginous (XIV9′i) when mature; margin acute; first incurved, turning wavy to up-lifted; rarely dehiscent; non-striate; peeling 1/6–1/5 towards the centre; with a red tinge of Scarlet Red (I3), Scarlet (I5) to Spectrum Red (I1). Lamellae adnate to adnexed, narrow, 2–4 mm in height at midway point of pileus radius, brittle, infrequently forked near the stipe, interveined, white to pale cream, initially White (LIII), turning Light Buff (XV17′f) to Massicot Yellow (XVI21′f) with age; edge even, acute, 10–15 pieces at 1 cm near the pileus margin, lamellulae rare. Stipe central; 57–98 × 8–14 mm; cylindrical to subcylindrical; sometimes slightly tapered towards the base; annulus absent; smooth to longitudinally rugulose; most of the surface red with Light Jasper Red (XIII3′b), Eugenia Red (XIII1′) to Coral Red (XIII5′), except for a pinkish tinge of Coral Pink (XIII5′d) to Jasper Pink (XIII3′d) towards the top; stuffed to farctate first, fistulous to hollow when mature. Context 2–4 mm thick at pileus centre, brittle, initially White (LIII), unchanging when bruised, becoming pale yellowish tinge of Cream Color (XVI19′f) to Naphthalene Yellow (XVI23′f) with age, taste more or less acrid, odour indistinct. Spore print pale cream (IIa–IIb).

*Basidiospores* (200/4/4) (5.8–) 6.2–9.3 (–9.8) × (4.6–) 5.0–8.1 (–7.8) μm, Q = (1.06–) 1.09–1.38 (–1.35) (Q = 1.20 ± 0.08), 7.6 × 6.2 μm on average, subglobose to broadly ellipsoid, occasionally ellipsoid, ornamentations composed of mostly tuberculate warts 0.3–0.9 μm in height, moderately distant to dense ((4–)6–7 in a 3 μm diam. circle), often connected by fine lines, often fused in pairs and short chains (1–3(–5) in the circle), line connections dispersed (0–1 in the circle); suprahilar area amyloid and distinct. Basidia 35–53 × 8–13 μm, subclavate to clavate, occasionally subcylindrical, four-spored, hyaline; sterigmata to 6–9 μm long, mostly straight. *Cheilocystidia* 44–73 × 6–11 μm, cylindrical to subcylindrical, rarely clavate to fusiform, apex obtuse, contents crystalline, black in SV. *Pleurocystidia* dispersed to moderately numerous, ca. 500–1100 mm^2^, 48–76 × 7–10 μm, clavate to subclavate, at times subcylindrical, apex obtuse, contents crystalline, partly dense, black in SV. Subhymenium 60–80 μm thick, cellular, partly pseudoparenchymatous, composed of inflated subglobose to ellipsoid, rarely elongated hyphae cells 7–10 μm in width. *Pileipellis* two-layered, distinctly differentiated from sphaerocytes in context. *Suprapellis* 100–150 μm thick, primordial hyphae with acid-resistant encrustations absent, an ixotrichoderm at pileus centre, composed of gelatinised, erect, oblique to repent hyphae, terminal cells cylindrical, 20–40 × 4–11 μm, apex obtuse, rarely tapered or ventricose, subapical cells cylindrical to inflated, rarely branched, 15–30 × 4–12 μm, an ixotrichoderm at pileus margin, composed of mostly repent to ascending elements, terminal cells 12–35 × 3–7 μm, cylindrical, obtuse at apex. *Pileocystidia* numerous in suprapellis, sometimes fasciculate at pileus centre, cylindrical, multi-septate, long, 70–150 × 7–11 μm, apex obtuse, contents dense, granulate, black in SV. Subpellis composed of loosely interlaced, mostly repent, cylindrical to ellipsoid hyphal cells 3–9 μm in width, longer hymenial cystidia up to 90 μm in length, and narrower pileocystidia 4–8.5 μm in width.

**Habitat and distribution:** Single or scattered in coniferous and broad-leaved mixed forests.

**Specimens examined:** China, Xizang Autonomous Region, Nyingchi City, Mainling County, Nanyi Valley, in mixed forest of *Cyclobalanopsis xizangensis*, *C. oxyodon*, *Picea likiangensis* var. *linzhiensis* and *Quercus lodicosa*; 29°9′28.0′′N, 94°12′31.9′′E; S.Y. Su, X.M. Bai, G.J. Li; 20 September 2015; ZRL20152381 (HMAS281960); ibid, 29°9′16.7′′N, 94°12′31.9′′E, ZRL20152382 (HMAS281961); ibid, 29°9′12.4′′N, 94°12′30.1′′E, ZRL20152395 (HMAS281962); ibid, 29°9′17.8′′N, 94°12′34.7′′E, ZRL20152371 (HMAS281958).

**Note:** The morphological character of the Xizang collections of this study are similar to that of Sichuan [25], except for the latter has narrower, mostly one-celled pileocystidia (3.8–)4.7–6.2–7.6(–9.5) μm in width. This species is reminiscent of *R. torulosa* and *R. thindii* in *R*. subsection *Sardoninae* due to its rosy red pileus and stipes, as well as coniferous-tree habitat. *Russula thindii* can be distinguished from its slightly bitter-tasting context, longer pleurocystidia of 55–123 × 12–15 μm with appendiculate apices, and shorter and narrower pileocystidia of 27–50 × 5–7 μm [23]; *R. torulosa* differs in its larger basidiomata up to 90 μm in pileus diameter, basidiospores with lower ornamentations up to 0.6 μm in height, long hymenial cystidia up to 150 μm in length [14,15]. For the other species close to the ITS phylogeny in Figure 1, *R. americana* is different from *R. roseola* in its larger basidiospores of 9–11.5 in diam. and habit of *Abies* forest [18]; *R. fuscorubroides* can be distinguished by its dark-purple- or brownish-violet-tinged pileus surface, lamellae later turning coarsely ochraceous and rarely greenish, and appendiculate hymenial cystidia apex [56]; *R. queletii* can be distinguished from the new species through its gooseberry-purple, wine-purple, and dark purple with brown tinges on its pilus surface, the fruity smell of its context, longer basidia of up to 60 μm in length, and hymenial cystidia attenuated to an acute, obtuse or finely capitate apexes [14,15]; *R. salishensis* differs in its occasional yellow-to-brown splotches on its pileus surface; fruity context odour; or, reminiscent of *Pelargonium*, subglobose to broadly ellipsoid basidiospores and narrower pileocystidia of up to 7.5 μm in width [20].

***Russula rufa* G.J. Li,** & **T.Z. Wei, sp. nov.** (Figure 4h and Figure 13).


**Fungal Names: FN 571250**


**Etymology:** Referring to the mainly reddish-tinged pileus.

**Holotype:** China, Xizang Autonomous Region, Qamdo City, Mangkam County, in coniferous forest of *Picea likiangensis* var. *linzhiensis*, *Pinus armandii* and *Pinus densata*; T.Z Wei, Z.X. Wu, L. Yang, H.D. Zheng, X.C. Wang; 16 August 2016; 7005 (HMAS277048).

**Diagnosis:** Basidiomata small to medium sized. Pileus mainly reddish tinged, often intermixed with colours of brown, green and purple, slightly glutinous when wet, margin indistinctly striate. Lamellae adnate to adnexed, white to cream, edge even, 7–14 blades at 1 cm near the pileus margin, lamellulae rare. Stipe 42–63 × 7–15 mm, subcylindrical to cylindrical, pale pinkish tinged. Context white to pale ocher, unchanging when bruised. Fragile, taste acrid, odour indistinct. Spore print pale cream. *Basidiospores* (5.4–) 6.1–11.1 (–12.1) × (4.4–) 5.1–9.0 (–10.0) μm, 7.7 × 6.4 μm in average, subglobose, broadly ellipsoid to ellipsoid, rarely globose, ornamentations 0.4–1.0 μm in height, reticulate, occasionally to frequently fused in pairs and short lines. *Basidia* 33–47 × 8–13 μm, cylindrical, subcylindrical to subclavate. *Cheilocystidia* rare, 43–70 × 8–12 μm, fusiform to clavate, apex lanceolate. *Pleurocystidia* 46–83 × 8–13 μm, fusiform to clavate, sometimes subcylindrical, apex lanceolate to papilliform. *Pileipellis* contains two layers, *Suprapellis* a trichoderm at pileus centre, partly an ixotrichoderm at the margin. *Pileocystidia* present, abundant in pileus centre, septate. Habitat in coniferous forests.

**Description:** Basidiomata small to medium sized. Pileus 34–56 mm in diam.; first pulvinate, hemispheric to convex, turning plano-convex to planate when mature; often shallowly depressed at centre, even infundibulum; mainly reddish tinged, sometimes intermixed with brownish, greenish and purplish tinges; Daphne Red (XXXVIII69′′b), Eupatorium Purple (XXXVIII67′′) to Tourmaline Pink (XXXVIII67′′b), rarely Livid Brown (XXXIX1′′′), Purplish Vinaceous (XXXIX1′′′b) and Deep Brownish Vinaceous (XXXIX5′′′), becoming Dragon’s Blood Red (XIII5′i), Dark Olive Buff (XL21′′′), and Vinaceous Tawny (XXVIII11′′) when mature; sometimes turning Ecru-Olive (XXX21′′i), Light Yellowish Olive (XXX23′′i) to Olive-Ocher (XXX21′′); dull, slightly glutinous when wet. margin acute to subacute, initially incurved, then flat, wavy and uplifted at last, infrequently cracked, indistinctly striated 1/6–1/4 from the edge inwards, peeling 1/6–1/4 of the radius with pale reddish to pinkish tinges of Pinkish Vinaceous (XXVII5′′d), Vinaceous (XXVII1′′d), and Corinthian Pink (XXXVII3′′d). Lamellae initially adnate, adnexed when mature, 2–4 mm in height at the mid-point of the pileus radius; White (LIII), unchanging when injured, occasionally becoming cream tinge of Ivory Yellow (XXX21′′f), Massicot Yellow (XVI21′f), and Cream Color (XVI19′f); brittle, occasionally forking near the stipe, slightly interveined; edge even, narrowing towards the margin, 7–14 blades at 1 cm near the pileus margin, lamellulae a few. Stipe central, 42–63 × 7–15 mm, subcylindrical to cylindrical, rarely subclavate, slightly ventricose towards the base, annulus absent, smooth when young, turning longitudinally rugulose at last, most of the surface pale pinkish tinged; Livid Pink (XXXVIII3′′f), Pale Persian Lilac (XXXVIII69′′f) to Pale Laelia Pink (XXXVIII69′′f), except for a partial White (LIII) towards the pileus; stuffed first, turning fistulous to hollow when mature. Context 2–3 mm thick at pileus centre, White (LIII), unchanging when bruised, sometimes slowly becoming pale ochre tinge of Cream Buff (XXX19′f), Aniline Yellow (IV19i) and Yellow Ocher (XV17′) with age; fragile, taste acrid, odour indistinct. Spore print pale cream (Romagnesi IIa–IIb). 

*Basidiospores* (200/4/4) (5.4–) 6.1–11.1 (–12.1) × (4.4–) 5.1–9.0 (–10.0) μm, Q = (1.03–) 1.07–1.35 (–1.40) (Q = 1.20 ± 0.09), 7.7 × 6.4 μm on average, subglobose, broadly ellipsoid to ellipsoid, rarely globose, ornamentations amyloid, composed of blunt, cylindrical, subcylincrical to subconical warts 0.4–1.0 μm in height, moderately distant to dense ((4–)6–8 in a 3 μm diam. circle), reticulate when young, interconnections turning infrequent with age, isolated, occasionally to frequently fused in pairs and short lines (0–2(–3)fusions in the circle), line connections dispersed (0–1 in the circle); suprahilar area amyloid, raised and distinct. *Basidia* 33–47 × 8–13 μm, cylindrical, subcylindrical to subclavate, at times clavate, four-spored, rarely two-spored, hyaline; sterigmata 4–7 μm in length, mostly straight, slightly curly. *Cheilocystidia* rare, 43–70 × 8–12 μm, fusiform to clavate, apex lanceolate, contents granular, black in SV. *Pleurocystidia* widely dispersed, ca. 100–200/mm^2^, 46–83 × 8–13 μm, fusiform to clavate, sometimes subcylindrical, apex lanceolate to papilliform, sometimes obtuse, contents crystalline to granular, partly sparse, black in SV. Subhymenium ca. 30 μm thick, composed of pseudoparenchymatous cells 8–13 μm in width. *Pileipellis* contains two layers, unambiguously distinct from the underlying sphaerocytes in context. *Suprapellis* a trichoderm 80–120 μm thick at pileus centre, composed of repent to oblique, cylindrical, hyaline hyphae, primordial hyphae absent, terminal cells 13–25 × 3–5 μm, apex obtuse, rarely ventricose or tapered, subapical cells cylindrical, rarely bifurcated, inflated to ellipsoid, 3–7 μm in width; suprapellis partly an ixotrichoderm in pileus margin, composed of gelatinized, horizontal to tilted elements, terminal cells 10–20 × 3–5 μm, cylindrical, apex obtuse, subapical cells subcylindrical, at times more or less inflated, up to 8 μm in width. Pileocystidia present in both of suprapellis and subpellis, abundant in pileus centre, 40–80 × 7–14 μm, cylindrical, often 1–3 septate, rarely multi-septate, apex obtuse, sometimes ventricose to tapered in pileus centre, contents granulate, mostly dense, blackish grey in SV. Subpellis a cutis, composed of mostly repent to irregularly interweaving, cylindrical to inflated, septate hyline hyphal cells 3–10 μm wide. Clamp connections not observed in all tissues.

**Habitat and distribution:** Single or scattered in coniferous forests.

**The rest of specimens examined:** China, Xizang Autonomous Region, Nyingchi City, Bomi County, Zhuolong Valley, in coniferous forest of *Picea likiangensis* var. *linzhiensis*, *Pinus armandii* and *Pinus densata*; T.Z Wei, Z.X. Wu, L. Yang, H.D. Zheng, X.C. Wang; 22 September 2016; 585 (HMAS 278797); ibid, 570-1 (HMAS278787); ibid, 560 (HMAS278779); ibid; Nyingchi County, 26 September 2016, 635 (HMAS281062); Sichuan Province, Aba Tibetan and Qiang Autonomous Region, Xiaojin County, in coniferous forest; T.Z. Wei, L.H. Sun, Z.X. Wu, R.C. Zhang; 11 August 2016; 5675 (HMAS276972).

**Notes:** This species could be confused with *R. cavipes*, *R. queletii*, *R. roseola*, *R. salishensis* and *R. sardonia* because of pileus-pinkish-red, wine-red, purplish-red, often intermixed with green, yellow-to-brown splotches, stipes with a pink flush, acrid context taste, and a habitat of coniferous forest [14,15,20]. However, *R. cavipes* was described as having a strong context odour of *Pelargonium* and laudanum, higher basidiospore ornamentations of up to 1.6 μm in height, longer and wider hymenial cysitidia of 70–140 × 9–18, and narrower pileocystidia of 7.5–10 μm in width [14,15]. *Russula queletii* can be distinguished from *R. rufa* by the presence of an obviously fruity context odour, rarely greenish lamellae, longer and wider hymenial cystidia of 55–130(–150) × 8–16 μm, and narrower pileipellis hyphae of 2.5–4 μm in width [14,15]. *Russula roseola* is differentiated from *R. rufa* by its numerous pleurocystidia (1000–1400/mm^2^); and mostly one-celled, narrower pileocystidia (3.8–)4.7–7.6(–9.5) μm in width [25]. *Russula salishensis* can be distinguished from its fruity or sometimes *Pelargonium* context smell, white stipes with faint-pink flush, not highly amyloid basidiospores suprahilar spots, subreticulate spore ornamentations, and narrower pileocystidia of up to 7.5 μm in width [20]. *Russula sardonia* is different from *R. rufa* through its having lower basidiospore ornamentations of up to 0.7 μm in height, longer hymenial cysitidia of up to 200 μm in length, and narrower pileocystidia of 3–8 μm in width [14,15].

## 4. Discussion

Although ITS phylogeny analyses have been regarded as one of the standardized methods in the delimitation and clarification of closely related *Russula* species [57], accurate macro- and microscopic morphology based on statistics and symbiotic plant information in detail are essentials in taxonomy [2,58]. The preference of ITS phylogenetic topology may lead to inaccurate infrageneric classifications and the oversight of existing species. *Russula queletii* complex 1 and the rhodochroa-subsanguinaria complex in ITS phylogeny are representative examples (Figure 1). Multi-locus phylogenetic analyses have become the preferred technique for revealing ties of consanguinity and cryptic species from *Russula* genus in recent years [57,59]. The lowest amplification success rate in this study are *rpb1* and *tef-1α* regions (ca. 57%). All of these four species, including *R. begonia*, *R. photinia*, *R. rhodochroa* and *R. rufa*, are supported by the morphological and multi-locus phylogenetic evidence of this study. This study also demonstrates wider distributions of *R. gracillima*, *R. leucomarginata*, and *R. roseola*. The seven species detailed in this study can be distinguished from the others in morphology.

The *Russula* subsection *Sardoninae* members described and illustrated in this study have mostly narrow, erect to synclinal, intricate, rarely branched terminal cells in pileipellis [14]. There are more inflated terminal elements in the pileipellis of *R. begonia*. Pileocystidia in suprapellis are always cylindrical, and often contain 1–3 septa. *Russula rufa* has wider pileocystidia compared with other species. The basidiospore ornamentation comparisons show that *R. roseola* has more interconnected warts and ridges, while *R. rhodochroa* and *R. rufa* have more isolated warts.

Some new Asian species of the *R*. subsection *Sardoninae* of this study may have been identified as European members such as *R. exalbicans*, *R. gracillima*, *R. luteotacta*, *R. persicina*, *R. queletii*, *R. sardonia* and *R. sanguinea* in morphology [60,61]. Intensive observations and ITS phylogenetic analyses indicated that some Chinese specimens have both similarities and differences [17,27,62,63,64]. Combined support of morphology, ITS and multi-locus phylogenies and broad-leaved forest habitats suggest *R. gracillima* have certain possible distributions in both Europe and North China (Figure 1 and Figure 2). This is because of the continuously distributed symbiotic host plants Betulaceae and Fagaceae members in the Holarctic region [65]. Recent phylogenetic analyses have not yet shown any support for the existence of *R. luteotacta* and *R. sardonia* in China. Certain degrees of ITS sequence differences (ca. 1–2%) have been detected in *R. exalbicans*, *R. queletii* and *R. sanguinea* collections from different continents [17,27,62,63,64]. Multi-locus phylogenetic analyses of Chen et al. [25] and this study further revealed more Asian discoveries in *R.* subsection *Sardoninae*: specimens of “*R. sanguinea*” are *R. leucomarginata*, *R. rhodochroa*, *R. roseola*, and *R. subsanguinaria*; “*R. queletii*” samplings are *R. rufa*; specimens of “*R. persicina*” are *R. begonia*; a portion of “*R. gracillima*” specimens are *R. photinia*. Whether these Chinese collections of “*R. exalbicans*” are cryptic species or intraspecific geographical populations still needs further analyses. Several misidentifications in GenBank were detected in this work, such as North American samplings identified as *R. fragilis* of *R*. subsection *Russula*.

The phylogenetic topology ITS region of this study (Figure 1) generally corresponds with that of Chen et al. [25]. Clade F corresponds to the not highly supported branch composed of *R. americana*, *R. fuscorubroides*, *R. leucomarginata*, *R. pseudopelargonia*, *R. rhodocephala*, *R. roseola*, *R. sanguinea* (as “*R. sanguinaria*”), *R. sardonia*, *R. subsanguinaria*, and *R. thindii* in Figure 1 of Chen et al. [25]. Clade G matches the strongly supported branch of *R. exalbicans*, *R. gracillima* and *R. persicina* in Figure 1 of Chen et al. [25]. Similar correspondences are also present in multi-locus phylogenetic analyses of these two studies. Still, there are some topological differences between ITS and multi-locus phylogenies, e.g., *R. luteotacta* clusters with members of clade G in Figure 1 (Figure 2). There are a number of species in Figure 1 that still need further multi-locus analyses for a lack of reference sequences other than ITS. The representatives of these species are *R. choptae*, *R. renidens*, *R. rhodopus*, *R. suecica*, and *R. vinosoflavescens*. Habitat features can be detected from our analyses, which show that members of Clade F grow in coniferous forest, and those of Clade G live in broad-leaved forest. The taxonomic ideology of Sarnari [14] on *R.* subsection *Sardoninae* has been partially supported by molecular phylogeny. Thus, concrete symbiotic tree species are important in field work. The identification of ectomycorrhiza is necessary in future *Russula* taxonomy, especially for those samplings from mixed forest. Four series, *Exalbicans*, *Sardonia*, *Persicina* and *Sanguinea*, were proposed according to the criteria of pileus colour and forest types in Sarnari [14]. Chen et al. [25] and our analyses show a polyphyly of the *R.* subsection *Sardoninae* members with similar pileus colours in molecular phylogenetic topologies, even for those species with the same habitat. This work also provides further evidence of intraspecific pileus colour variations in *R.* subsection *Sardoninae* (Figure 3 and Figure 6). Therefore, it is error-prone to use pileus as the taxonomic standard in this subsection. Our work indicates the plant flora of a habitat can be used as an interspecific delimitation standard for those species with high ITS similarities. *Russula rhodochroa* and *R. subsanguinaria* can be distinguished by their habitats. As there is an obvious floristic difference between Northeast China and the Yunnan–Guizhou–Guangxi regions in the Sino-Japanese forest subkingdom [65]. Intensive analyses are needed for the Chinese record of *R. thindii* (RITF2712) collected from the Hubei Province of the Sino-Japanese forest subkingdom, because the type specimen of this taxon is reported in forest of the Sino-Himalayan floral subkingdom [23].

Although the context and lamellae of *R.* subsection *Sardoninae* members are more or less acrid in taste, *R. sanguinea* and *R. sardonia* of this group have been recorded as edible fungi in China [66,67]. This can be explained by the fact that maturation and dehydration often turn the basidiome context from acrid to insipid in taste. As the Chinese distribution of these two species has not been supported by molecular phylogeny, the edibility of *R.* subsection *Sardoninae* still needs further sampling and food safety analyses. Basidiomata of *R. rhodochroa* are occasionally collected and sold with other mild-tasting *Russulas* in southeastern region of Guizhou Province, according to our recent surveys. Red-capped *Russula* members in northeastern China are often regarded as inedible species with the local name “coffin lid”. As raw and incomplete cooked *R.* subsection *Sardoninae* members may cause gastrointestinal symptoms [68], it is highly risky to label these species as edible fungi.

## Figures and Tables

**Figure 1 jof-09-00199-f001:**
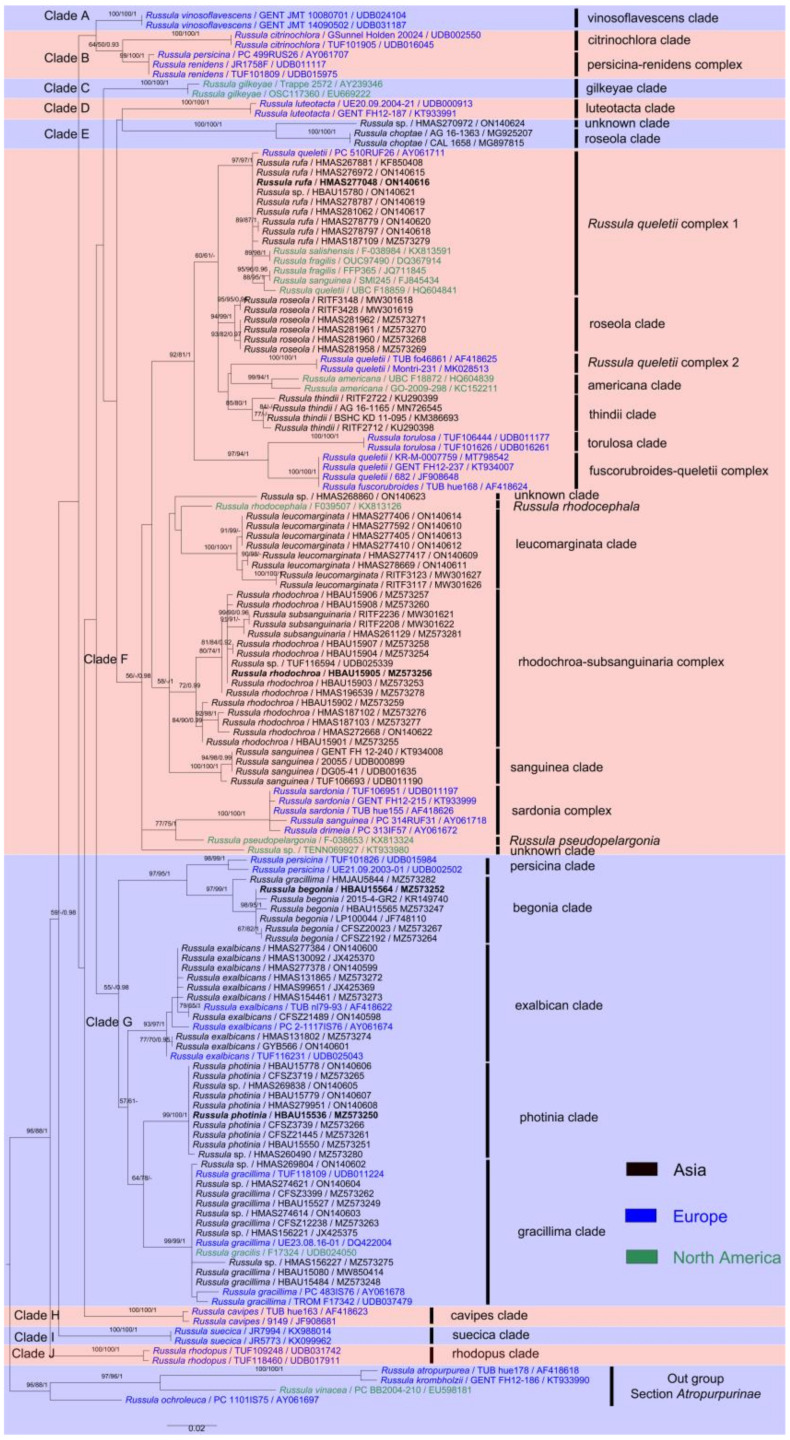
The ITS phylogenetic topology derived from maximum likelihood analysis. Bootstrap values (≥50%) of maximum likelihood (MLBS) and maximum parsimony (MPBS) analyses, together with posterior probability (PP) values (≥0.9) of Bayesian analysis, are presented above the branches as (MLBS/MPBS/PP). Holotypes of new species are shown in bold.

**Figure 2 jof-09-00199-f002:**
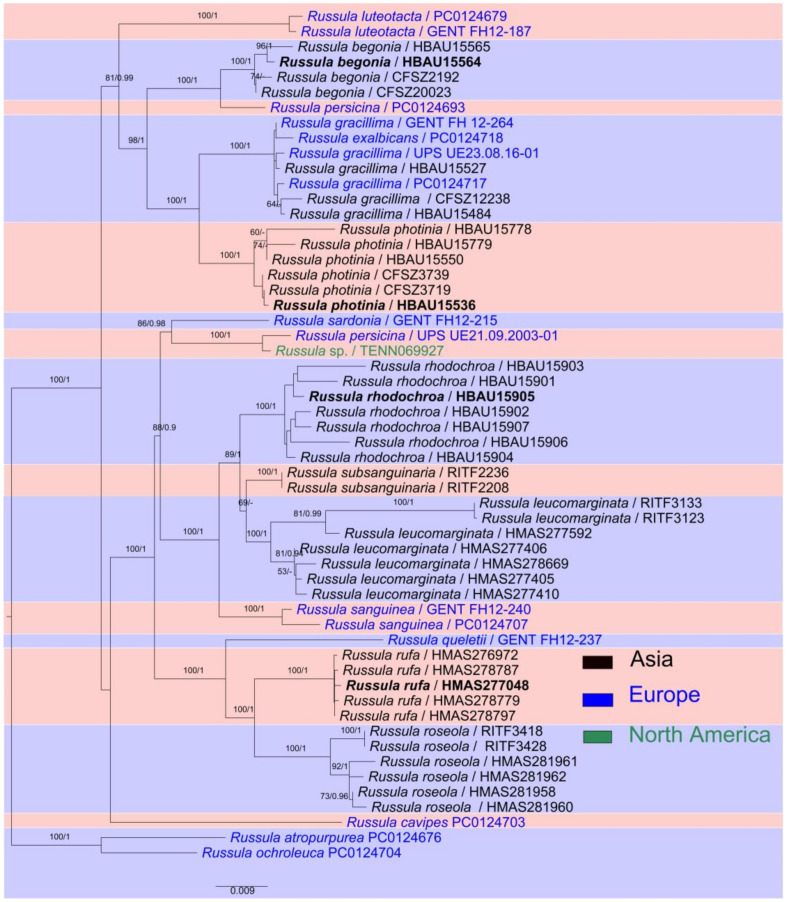
Multi-locus maximum likelihood phylogenetic tree of nLSU-mtSSU-*rpb1*-*rpb2*-*tef*-*1α* regions. Holotypes of the new species are shown in bold. MLBS (≥50%) and PP (≥0.9) values are presented above the nodes as (MLBS/PP).

**Figure 3 jof-09-00199-f003:**
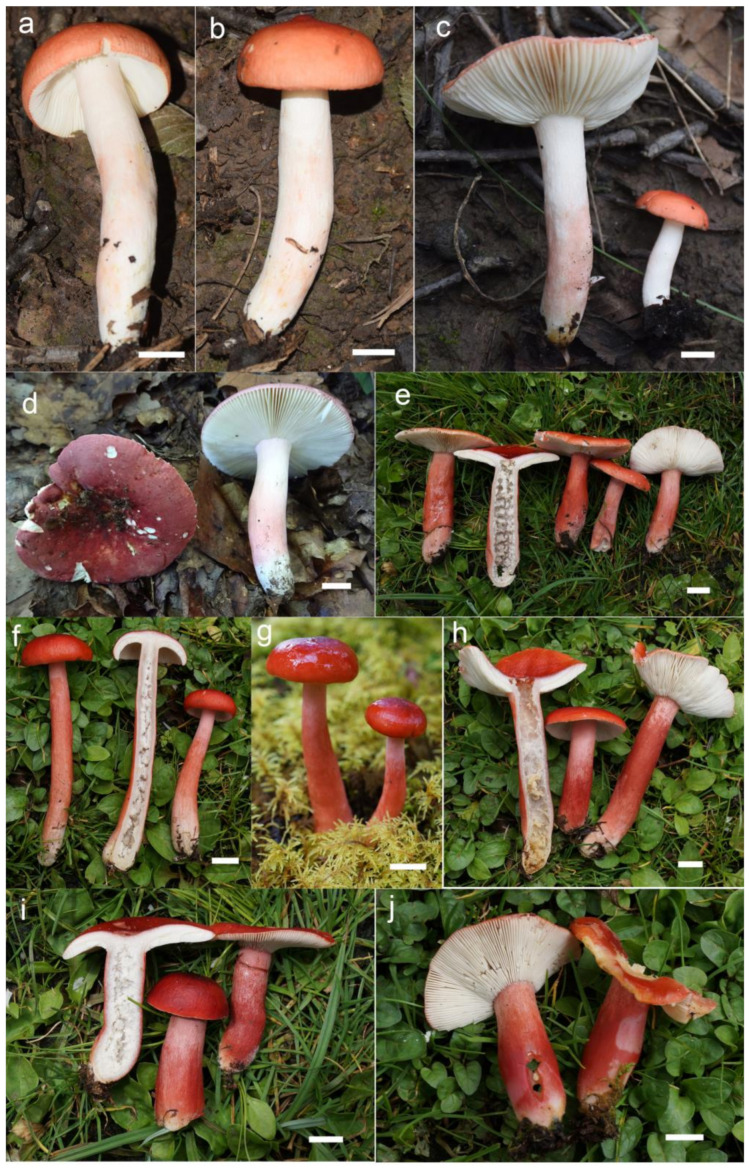
Basidiomata. (**a**–**d**) *Russula begonia*. (**e**–**j**) *R. roseola*. Bars: 10 mm.

**Figure 4 jof-09-00199-f004:**
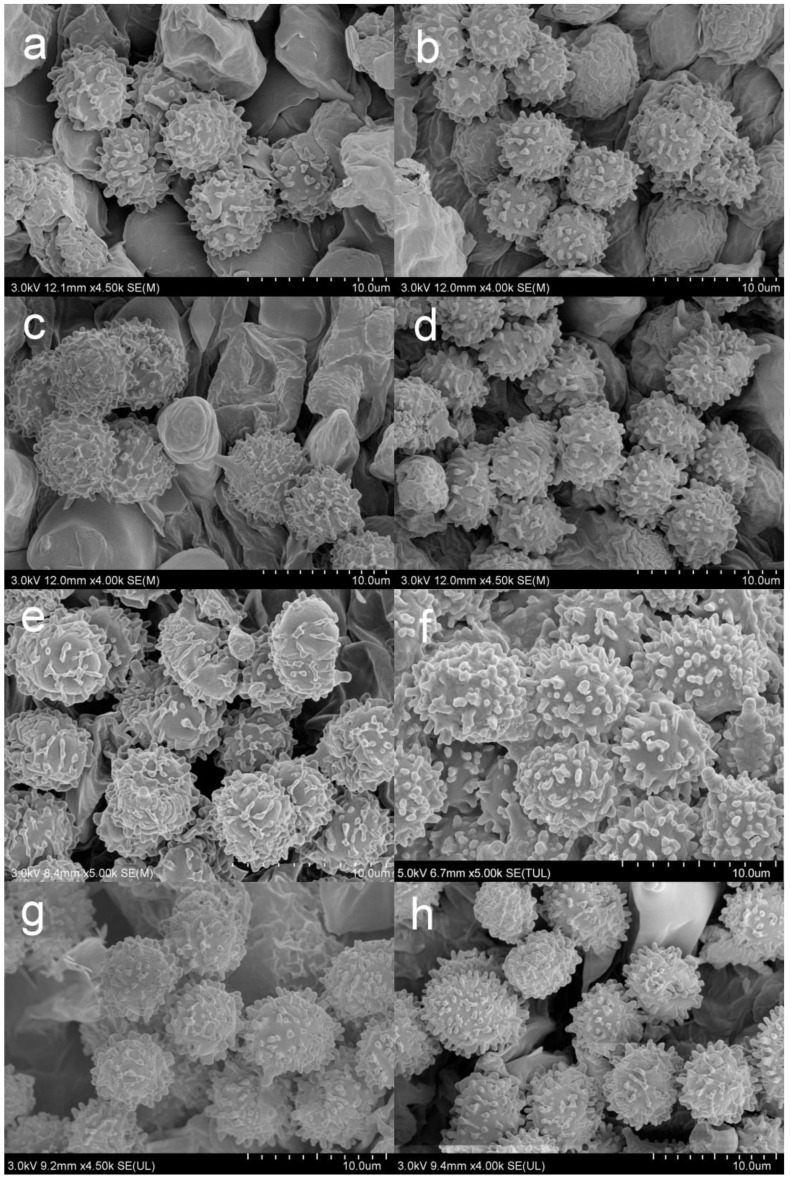
SEM photos of basidiospores. (**a**) *Russula begonia*. (**b**) *R. gracillima*. (**c**) *R. roseola*. (**d**,**e**) *R. photinia*. (**f**) *R. leucomarginata*. (**g**) *R. rhodochroa*. (**h**) *R. rufa*.

**Figure 5 jof-09-00199-f005:**
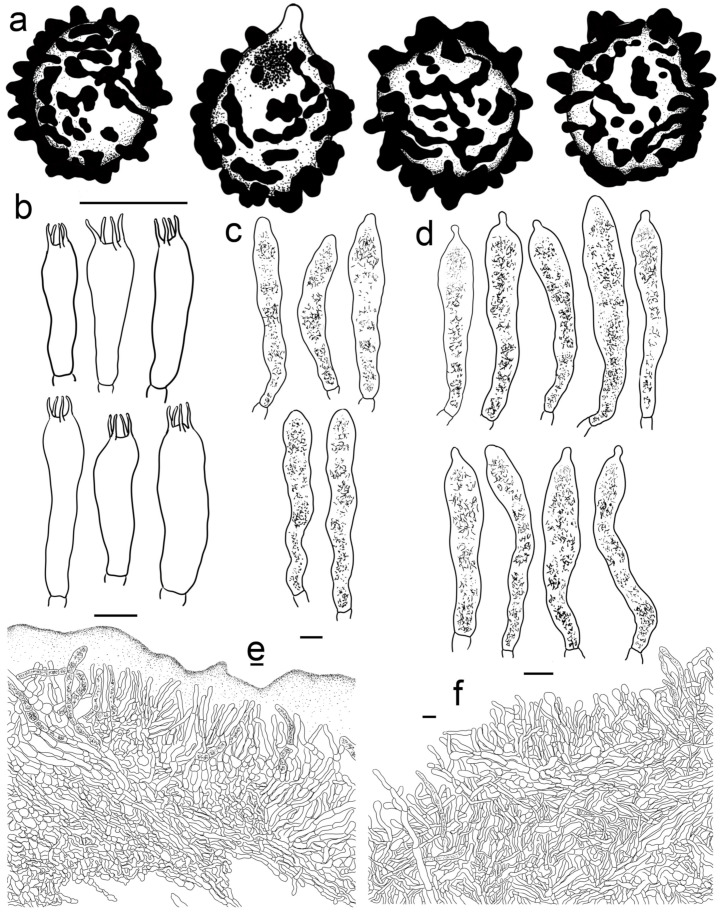
*Russula begonia* (HBAU15564, Holotype). (**a**) Basidiospores. (**b**) Basidia. (**c**) Cheilocystidia. (**d**) Pleurocystidia. (**e**) Suprapellis and partial subpellis in pileus centre. (**f**) Suprapellis in pileus margin. Bars: (**a**) = 5 μm, (**b**–**f**) = 10 μm.

**Figure 6 jof-09-00199-f006:**
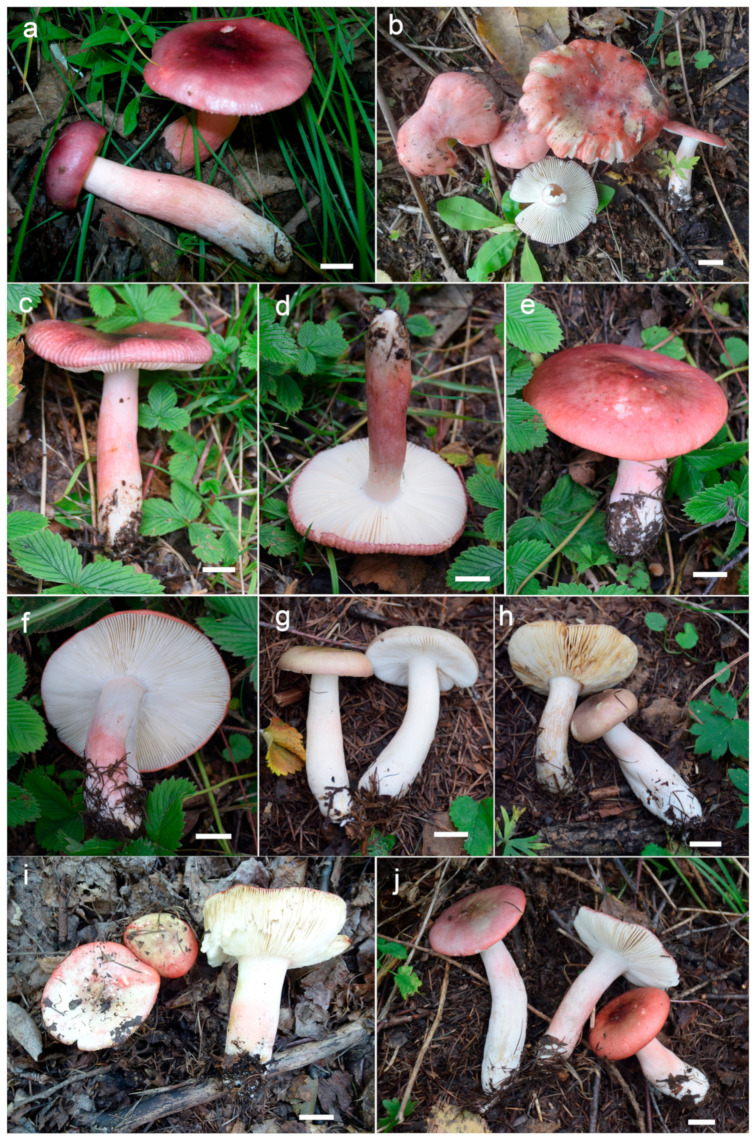
Basidiomata. (**a**–**d**,**g**,**h**) *Russula gracillma*. (**e**,**f**,**i**,**j**) *R. photinia*. Bars: 10 mm.

**Figure 7 jof-09-00199-f007:**
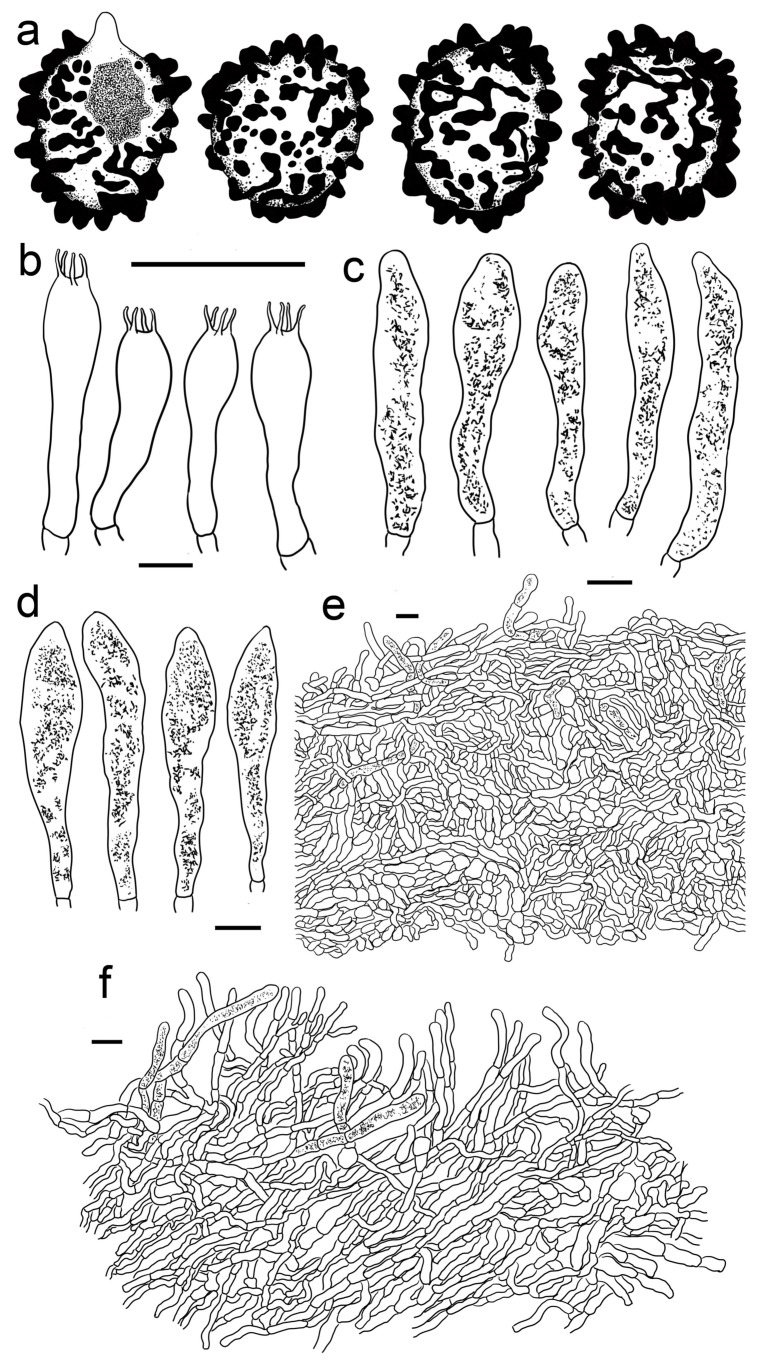
*Russula gracillima* (HBAU15527). (**a**) Basidiospores. (**b**) Basidia. (**c**) Cheilocystidia. (**d**) Pleurocystidia. (**e**) Suprapellis and partial subpellis in pileus centre. (**f**) Suprapellis in pileus margin. Bars: (**a**) = 5 μm, (**b**–**f**) = 10 μm.

**Figure 8 jof-09-00199-f008:**
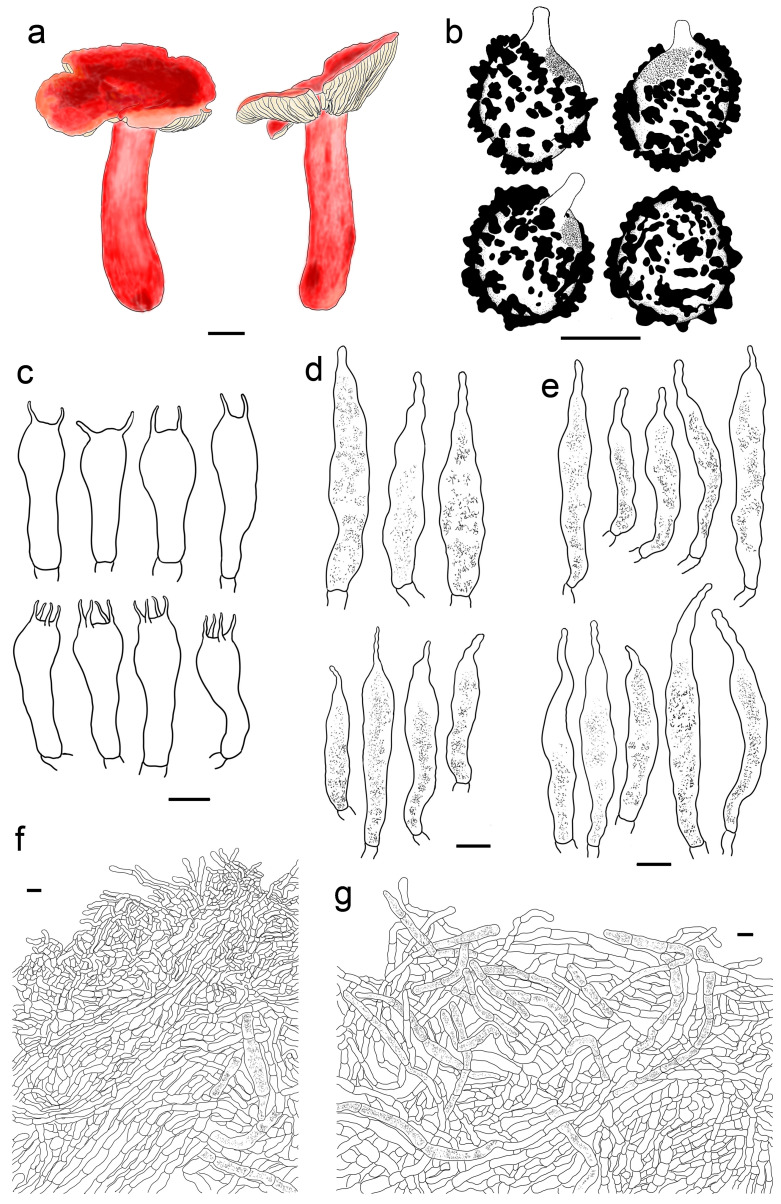
*Russula leucomarginata* (HMAS277410). (**a**) *Basidiomata*. (**b**) *Basidiospores*. (**c**) *Basidia*. (**d**) *Cheilocystidia*. (**e**) *Pleurocystidia*. (**f**) *Suprapellis* in pileus centre. (**g**) *Suprapellis* in pileus margin. Bars: (**a**) = 10 mm, (**b**) = 5 μm, (**c**–**g**) = 10 μm.

**Figure 9 jof-09-00199-f009:**
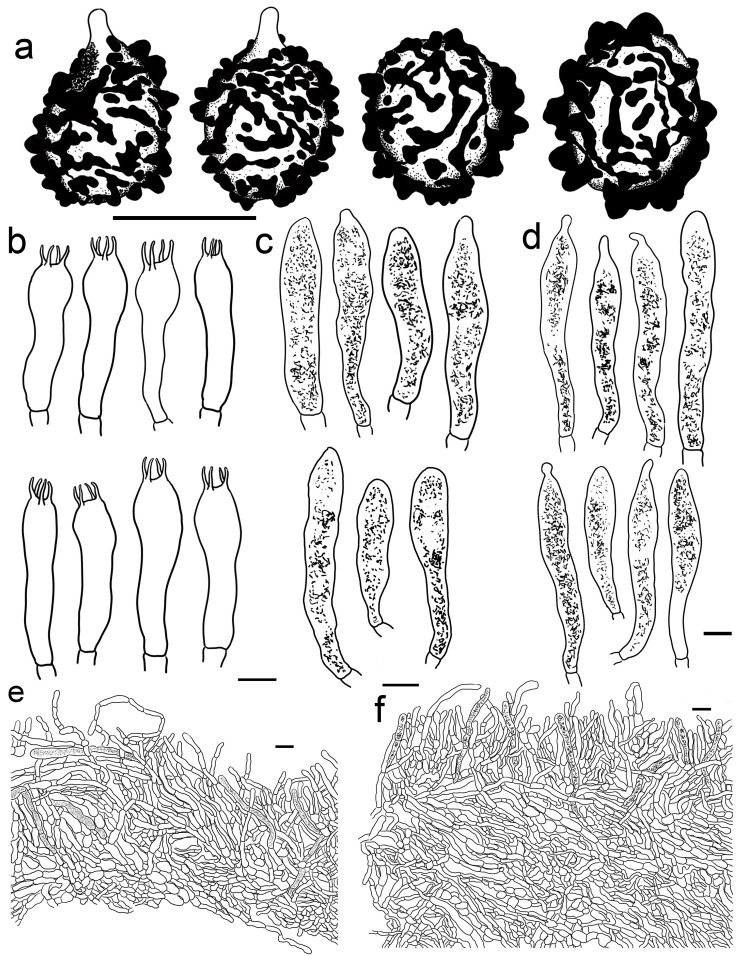
*Russula photinia* (HBAU15536, Holotype). (**a**) *Basidiospores*. (**b**) *Basidia*. (**c**) *Cheilocystidia*. (**d**) *Pleurocystidia*. (**e**) *Suprapellis* and partial subpellis in pileus centre. (**f**) *Suprapellis* and partial subpellis in pileus margin. Bars: (**a**) = 5 μm, (**b**–**f**) = 10 μm.

**Figure 10 jof-09-00199-f010:**
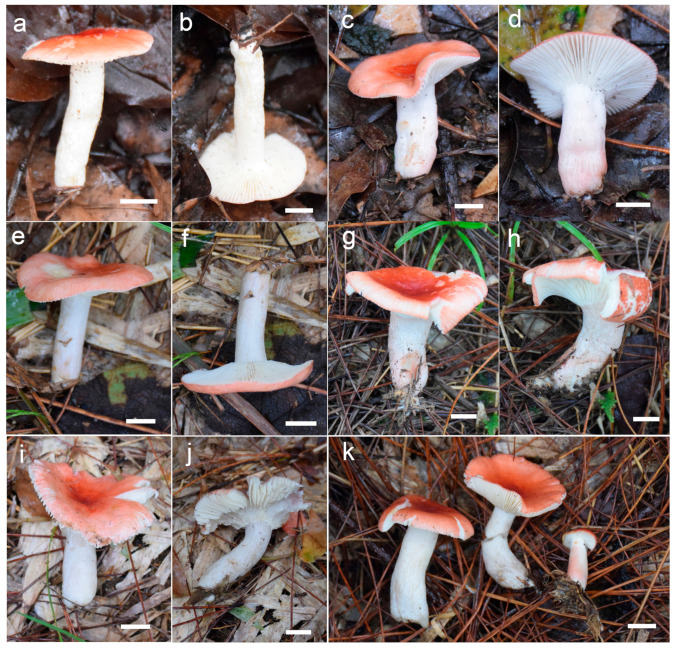
Basidiomata. (**a**–**k**) *Russula rhodochroa*. Bars: 10 mm.

**Figure 11 jof-09-00199-f011:**
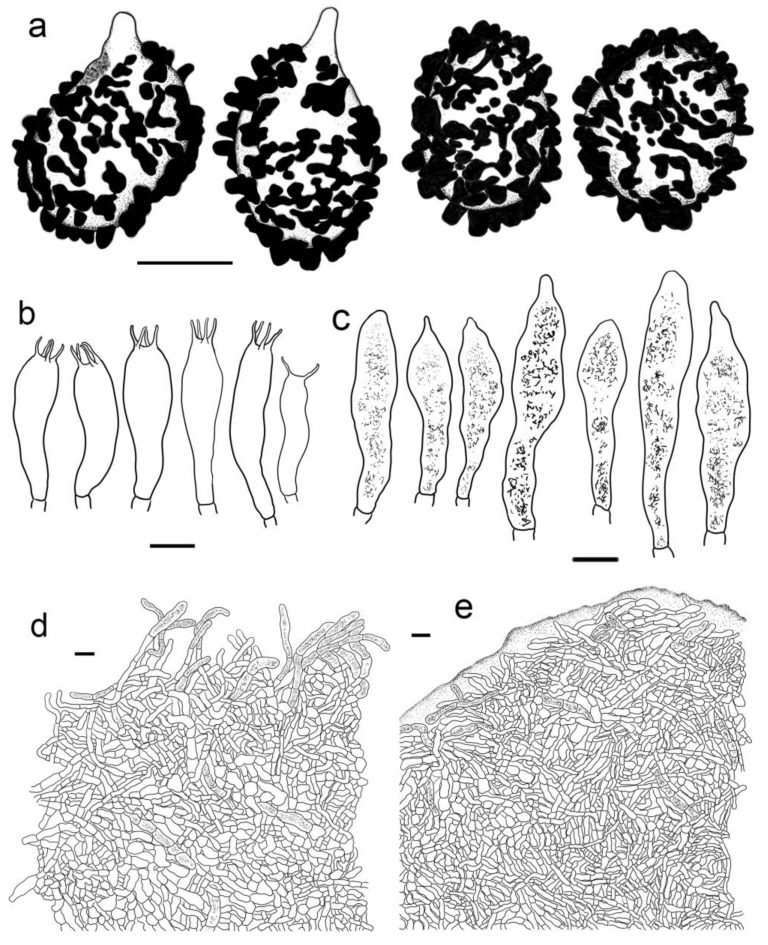
*Russula rhodochroa*. (HBAU15905, Holotype). (**a**) *Basidiospores*. (**b**) *Basidia*. (**c**) *Pleurocystidia*. (**d**) *Suprapellis* and partial subpellis in pileus centre. (**e**) *Suprapellis* and in pileus margin. Bars: (**a**) = 5 μm, (**b**–**e**) = 10 μm.

**Figure 12 jof-09-00199-f012:**
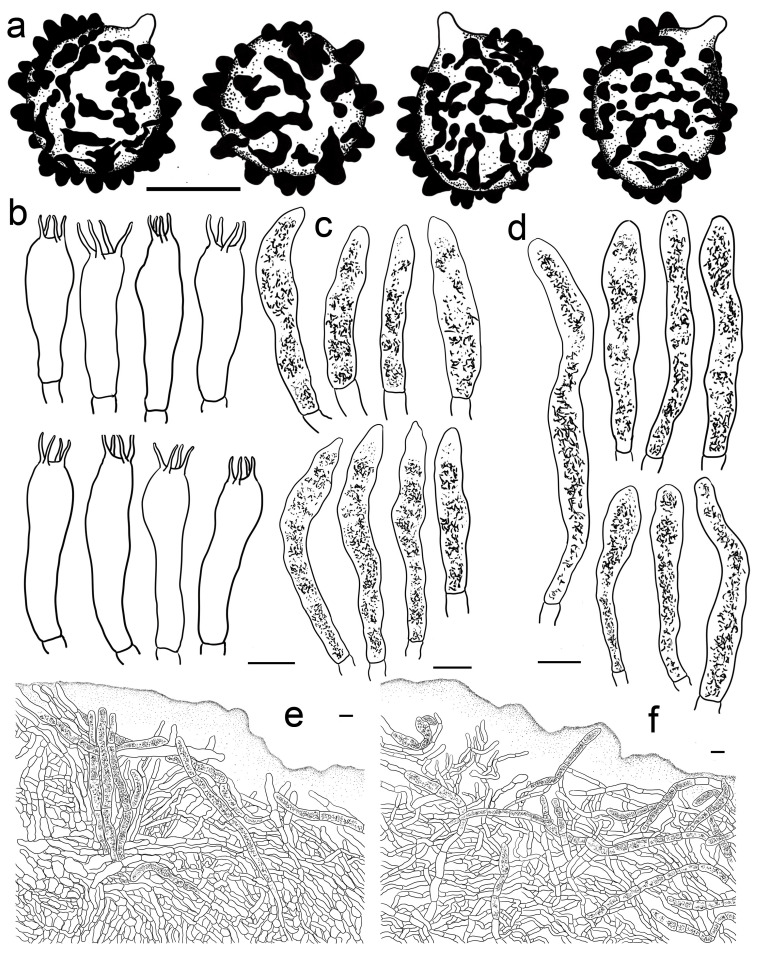
*Russula roseola*. (HMAS281958). (**a**) *Basidiospores*. (**b**) *Basidia*. (**c**) *Cheilocystidia*. (**d**) *Pleurocystidia*. (**e**) *Suprapellis* in pileus centre. (**f**) *Suprapellis* in pileus margin. Bars: (**a**) = 5 μm, (**b**–**f**) = 10 μm.

**Figure 13 jof-09-00199-f013:**
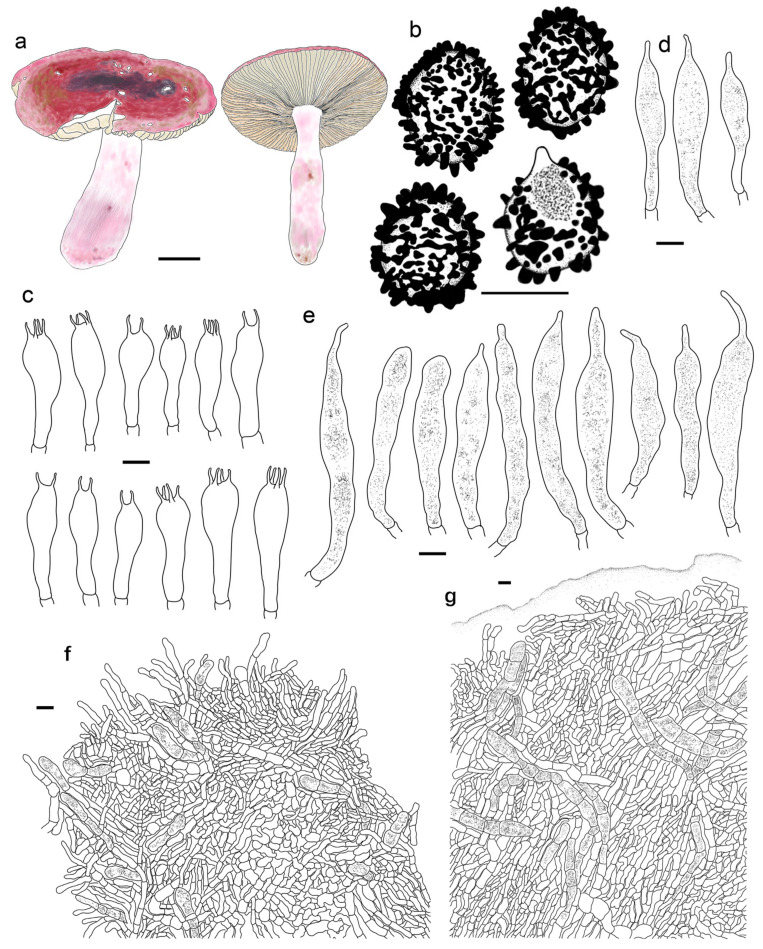
*Russula rufa*. (HMAS277048, Holotype). (**a**) *Basidiomata*. (**b**) *Basidiospores*. (**c**) *Basidia*. (**d**) *Cheilocystidia*. (**e**) *Pleurocystidia*. (**f**) *Suprapellis* and partial subpellis in pileus centre. (**g**) *Suprapellis* partial subpellis and in pileus margin. Bars: (**a**) = 10 mm, (**b**) = 5 μm, (**c**–**g**) = 10 μm.

## Data Availability

All sequence data are available in UNITE and NCBI GenBank following the accession numbers in the manuscript.

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
