# Peer review of "Four New Species of Russula Subsection Sardoninae from China"

_jof, 2023, doi:10.3390/jof9020199_

Round 1
Reviewer 1 Report
Dear authors,
The authors proposed four new species of Russula as well as provided evidence of three existing species in China. The manuscript is well-written. Phylogenetic analyses and morphological studies are done sufficiently. Results are logical and supported. Illustrations of characters of all species are beautiful. Only one point is that the drawings of pileipellis is rather small. Because the authors did not illustrate pileal elements separately, in a few pictures, it is not easy to distinguish pileocystidia clearly from other cells. In addition, it is preferable that diagnostic characters of proposed species should be concise and easy to understand. It should be a combination of unique characters of species.
Please find my comments in attached word file.

Author Response
Comments
- Page 2, line 63: revise to China during field fungal forays…?
Reply: The authors have rewritten this sentence. The detailed contents are highlighted in yellow (Lines 70-72).
2.Page 3, line 131: revise to All of sites in the matrix….?
Reply: The authors have revised this point following the reviewer’s suggestion (lines 143-144).
3.Diagnosis of proposed species should be concise. Please simplify it by designating only distinct characters
Reply: The diagnose parts of this manuscript part contain some basic morphological characters because some databanks will include the diagnose parts of from published papers. Descriptions will not be included. If only short and brief diagnoses were provided, it will cause some unnecessary inconvenience for databank users.
4.Page 13, line 297: revise to R. luteotacta?
Reply: The authors have revised this typo following the reviewer’s suggestion (line 314).
5.Page 13, line 302-303: authors mentioned spore size of R. exalbicans twice
Reply: The authors have revised this typo following the reviewer’s suggestion (line 318).
6.Drawings of pileipellis are rather small. In addition, can the authors indicate more clearly which cell is pileocystidia
Reply: The authors have checked the line drawings of pileipellis. The distinguishability of the drawings are enough to distinguish pileocystidia with contents.
7.There are some typos e.g. cheilocystidia in descriptions, subclavate (page 23, line 611), more or less acrid (page 26, line 700), distinct (page 26, line 707), ixotrichoderm (page 28, line 775)
Reply: The authors have checked the manuscript and revised the typos (line 283, 333, 365, 428, 463, 508, 528, 563, 614, 631, 664, 748, 796, 813, 853, 900), subclavate (line 625), more or less acrid (line 740), (line 747), ixotrichoderm (line 816, 863).
8.Page 17, line 418: delete “sometimes”?
Reply: The authors have revised this point following the reviewer’s suggestion (line 446).
9.Page 21, line 561-562: revise to The red form of this species?
Reply: The authors have revised this point following the reviewer’s suggestion (line 597).
- Page 27, line 755: delete “and”?
Reply: The authors have revised this point following the reviewer’s suggestion (Line 796).
11.Page 30, line 870: delete “higher”?
Reply: The authors have revised this point following the reviewer’s suggestion (Line 913).
12.Page 31, line 924: delete “and”?
Reply: The authors have revised this point following the reviewer’s suggestion (Line 970).
Reviewer 2 Report
In this manuscript, Li et al. reported four new species based on the morphological characteristics and multi-locus phylogeny.
This manuscript is well organized and written to provide the foundation for studying on the diversity and evolution. I have some concerns and suggestions listed below.
1)Line 26, Introduction section, I suggest to add some taxonomic history of Russula.
2)Line 47, ….known R. subsection, “R.” should be the whole name.
3)Line 398, Figure 8 f, delete the 10 μm in the picture 8 for the uniformity in the whole picture.
4)Line 681, I found that Russula roseola is the known species and reported recently, why do you descript it in this study again? Please explain it.
5)Line 737, Russula thindii can be written as R. thindii.
6)Line 870, “higher ”is right? I mean if it should be deleted. If no, please revised the expression.
7)Line 875, The R. subsection…, ‘R.’ should be written in the whole name.
8)Line 877, R.begonia should used in the whole name, while, Line 878, Russula rufa can be abbreviated as R. rufa.
9)Line 1000, there should have the “.” Before Nova Hedwigia.
10) In the reference section, I found some names of Journal are abbreviation, some are whole name, I suggest they should be uniform. Please check the whole references format.
Author Response
1) Line 26, Introduction section, I suggest to add some taxonomic history of Russula.
Reply: The authors have added the taxonomic history of Russula (line 38-44).
2)Line 47, ….known R. subsection, “R.” should be the whole name.
Reply: The authors have revised this point following the reviewer’s suggestion(line 54).
3)Line 398, Figure 8 f, delete the 10 μm in the picture 8 for the uniformity in the whole picture.
Reply: The authors have removed the scale bar following the reviewer’s suggestion (Figure 8).
4)Line 681, I found that Russula roseola is the known species and reported recently, why do you descript it in this study again? Please explain it.
Reply: Chen et al. (2022) described Russula roseola based on specimens described from Sichuan. The concept of a species is better to based on specimens from vast distributions than collections of only one site. There are some intraspecific morphological and phylogenetic variations among specimens from different regions. These differences are illustrated and discussed in Figures 1, 2 and 12, as well as note part of R. roseola. These illustrations and notes can be regarded as supplements of this species.
5)Line 737, Russula thindii can be written as R. thindii.
Reply: The abbreviations of genus name cannot be located as the first word of a sentence.
6)Line 870, “higher ”is right? I mean if it should be deleted. If no, please revised the expression.
Reply: The typo has been revised (line ).
7)Line 875, The R. subsection…, ‘R.’ should be written in the whole name.
Reply: The authors have revised this point following the reviewer’s instruction (line 913).
8)Line 877, R.begonia should used in the whole name, while, Line 878, Russula rufa can be abbreviated as R. rufa.
Reply: The genus and species names “Russula” and “Russula begonia” have been abbreviated in this manuscript several times earlier. The abbreviations of genus name of “R.” cannot be located as the first word of a sentence for “Russula rufa”.
9)Line 1000, there should have the “.” Before Nova Hedwigia.
Reply: This name of journal is in full spelling, thus no dot is needed.
10) In the reference section, I found some names of Journal are abbreviation, some are whole name, I suggest they should be uniform. Please check the whole references format.
Reply: The authors have checked the journal names. Necessary corrections are made and highlighted in yellow. There are some journals that did not have official abbreviations in the ISSN Center's List of Title Word Abbreviations or CAS's Core Journals List. So the authors kept the full spellings of these journal names.
Reviewer 3 Report
There are a few minor edits I have made in the manuscript.

Author Response
The authors have revised the manuscript following the reviewer’s comments. The revised contents are highlighted in yellow.
Concrete revisions corresponding with the reviwer can be seen in following lines (19, 20, 29, 72, 250, 302, 383, 481, 531, 580, 635, 681, 767, 817, 872).